# OPRIDE: Offline Preference-based Reinforcement Learning via In-Dataset Exploration

**Yiqin Yang**[1][*] **Hao Hu**[3][*] **Yihuan Mao**[2]**, Jin Zhang**[2]**, Chengjie Wu**[2]**, Yuhua Jiang**[2]**,**
**Xu Yang**[2]**, Runpeng Xie**[1]**, Yi Fan**[4][†] **Bo Liu**[5]**, Yang Gao**[2]**, Bo Xu**[1]**, Chongjie Zhang**[6]
[1]The Key Laboratory of Cognition and Decision Intelligence for Complex Systems,
 Institute of Automation, Chinese Academy of Sciences
[2]Tsinghua University [3]Moonshot AI [4]Amazon [5]University of Arizona
[6]Washington University in St. Louis
`yiqin.yang@ia.ac.cn`

## Abstract

Preference-based reinforcement learning (PbRL) can help avoid sophisticated reward designs and align better with human intentions, showing great promise in various real-world applications. However, obtaining human feedback for preferences can be expensive and time-consuming, which forms a strong barrier for PbRL. In this work, we address the problem of low query efficiency in offline PbRL, pinpointing two primary reasons: inefficient exploration and overoptimization of learned reward functions. In response to these challenges, we propose a novel algorithm, **O**ffline **P**b**R**L via **I**n-**D**ataset **E**xploration (OPRIDE), designed to enhance the query efficiency of offline PbRL. OPRIDE consists of two key features: a principled exploration strategy that maximizes the informativeness of the queries and a discount scheduling mechanism aimed at mitigating overoptimization of the learned reward functions. Through empirical evaluations, we demonstrate that OPRIDE significantly outperforms prior methods, achieving strong performance with notably fewer queries. Moreover, we provide theoretical guarantees of the algorithm's efficiency. Experimental results across various locomotion, manipulation, and navigation tasks underscore the efficacy and versatility of our approach.

## 1 Introduction

Reinforcement learning (RL) has proven effective across a range of sequential decision-making tasks, from mastering games like Go (Silver et al., 2016) to controlling complex systems such as robots (Ahn et al., 2022) and plasma reactors (Degrave et al., 2022). However, in many real-world applications, designing an appropriate reward function is a daunting challenge, as these tasks often involve objectives that are difficult to formalize with numerical rewards (Yang et al., 2021; 2023).

Preference-based RL (PbRL) (Akrour et al., 2012; Christiano et al., 2017) has emerged as a promising paradigm, leveraging human feedback in the form of pairwise preferences, which are inherently more interpretable yet still information-rich. This paradigm allows agents to learn from relative judgments rather than numerical reward signals, significantly reducing the complexity of reward design. Recent advancements in PbRL have illustrated its efficacy in enabling agents to learn novel behaviors (Christiano et al., 2017; Kim et al., 2023) and in achieving better alignment with human preferences (Ouyang et al., 2022; Yang et al., 2025), which are often difficult to encapsulate in a reward function. Despite these advantages, PbRL methods still face critical challenges, particularly in acquiring human feedback efficiently. Querying human preferences is both time-consuming and resource-intensive, limiting the scalability of PbRL in real-world applications.

To address this challenge, we propose **Offline PbRL via In-Dataset Exploration (OPRIDE)**, a novel algorithm designed to systematically enhance the query efficiency of offline PbRL, as depicted in Figure 1. OPRIDE introduces a principled exploration strategy that identifies the most informative

---

[*]These authors contributed equally.
[†]The work does not relate to their position at Amazon.

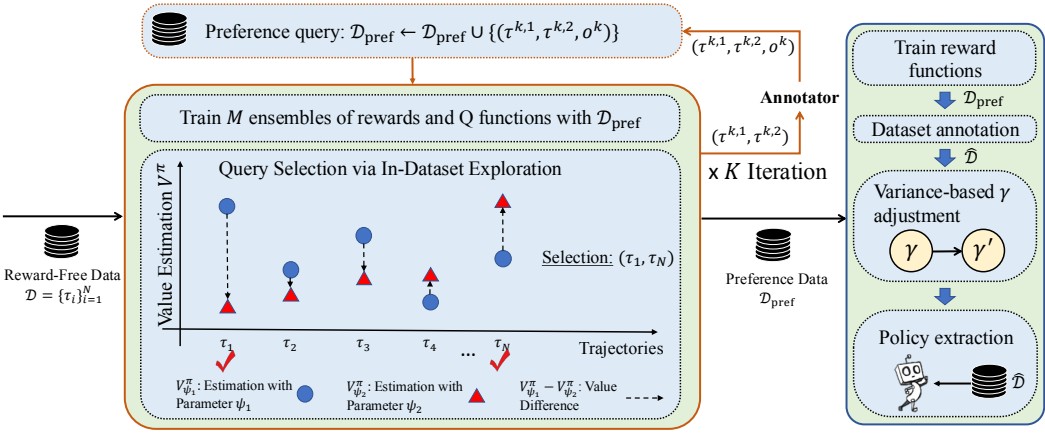

Figure 1: The procedure of OPRIDE consists of two phases. In the first offline phase, we select query based on exploration mechanism. The blue circles ● and red triangles ▲ represent the value estimation $V_{\psi_1}$ and $V_{\psi_2}$, respectively. In the second stage, we first learn an reward function based on the preference dataset and then annotate the reward-free dataset. Next, we adjust the discount factor to reduce the impact of noise in the reward learning.

queries by analyzing value differences between trajectories, ensuring that each query maximally contributes to learning the optimal policy. Additionally, to prevent overoptimization of the learned reward function (Gao et al., 2023; Zhu et al., 2024), particularly in regions with high uncertainty, we incorporate a discount factor scheduling mechanism that dynamically adjusts the discount based on the variance in the reward estimation. Based on the pessimistic property of the smaller discount factor, we can address the overestimation issue of the value function and, subsequently, a better policy performance and higher query efficiency.

Experimental evaluations on diverse locomotion and manipulation tasks, including AntMaze (Fu et al., 2020) and Meta-World (Yu et al., 2019), demonstrate the efficacy of our approach in achieving strong performance with significantly fewer queries compared to state-of-the-art baselines. Remarkably, our method achieves compelling results with as few as ten queries on Meta-World tasks, underscoring its efficiency and scalability. Furthermore, we provide theoretical insights into the efficiency of our algorithm, demonstrating that our exploration strategy is provably efficient under mild assumptions.

Our contributions are threefold: (1) We introduce OPRIDE, a novel offline PbRL algorithm that achieves superior query efficiency through in-dataset exploration; (2) We conduct extensive ablation studies that highlight the effectiveness of each component, providing insights into the factors driving query efficiency; and (3) We provide theoretical analyses establishing the provable efficiency of our algorithm involving a principled exploration strategy under mild assumptions.

### 1.1 RELATED WORK

**Preference-based RL.** Various methods have been proposed to leverage human preferences (Akrour et al., 2012; Ibarz et al., 2018) and have demonstrated success in tackling complex control tasks (Christiano et al., 2017; Lee et al., 2021) and in aligning large language models (Stiennon et al., 2020; Ouyang et al., 2022; Rafailov et al., 2023; 2024). In the realm of offline Preference-based Reinforcement Learning (PbRL), a benchmark including several baselines (e.g., disagreement based method) is introduced by OPRL (Shin et al., 2023), which selects queries based on disagreement between the reward models and is inefficient in determining the optimal policy. Kim et al. (2023) apply Transformer models to effectively capture preferences for better credit assignment. Kang et al. (2023) present a direct approach to learning policy based on preferences. A recent work by Lindner et al. (2021) proposes an information-directed query selection method for PbRL, using the Laplacian approximation and the Hessian matrix for posterior computation. In contrast, our method selects queries to maximize the information gain about the optimal policy rather than the reward function, ensuring higher query efficiency.

Our work is also related to recent two-stage offline PbRL methods that enhance reward learning or query design. CLARIFY (Mu et al.) employs contrastive learning to disambiguate preferences in noisy queries by refining trajectory representations, differing from our direct exploration of value differences within the dataset. LiRE (Choi et al., 2024) introduces listwise ranking to replace pairwise queries, improving feedback efficiency through sequential comparisons. Differently, we focus on maximizing policy-relevant information per query via value ensemble disagreement. Additionally, we note emerging single-stage paradigms that bypass explicit reward modeling: IPL (Hejna & Sadigh, 2023) and CPL (Hejna et al.) derive policies directly from preferences via inverse learning or contrastive objectives, while DPPO (An et al., 2023) optimizes policies using preference logits without reward intermediates. These methods represent an alternative direction, whereas OPRIDE retains the two-stage structure to leverage established offline RL algorithms.

In addition to empirical achievements, prior studies have also explored the theoretical aspects of PbRL. Pacchiano et al. (2021) propose a provable PbRL algorithm tailored for linear MDPs. Chen et al. (2022) extend this approach to scenarios where the Eluder dimension is finite. Zhan et al. (2023a) delve into the study of PbRL within an offline setting where a preference dataset is provided. Wang et al. (2023) propose an efficient randomized algorithm for PbRL in linear MDP and an efficient TS-based algorithm for nonlinear cases with finite Eluder dimensions. Sekhari et al. (2023) provides a PbRL algorithm with PAC guarantees. Novoseller et al. (2020) proposes the dueling posterior sampling algorithm that has an information-theoretic guarantee. Xu et al. (2020) provide a gap-dependent analysis for preference-based contextual bandit and imitation learning. Wu & Sun (2023) analyze the complexity of learning with utility-based preferences and general preferences.

## 2 PRELIMINARIES

We consider infinite-horizon Markov Decision Processes (MDPs), defined by the tuple $(\mathcal{S}, \mathcal{A}, \gamma, \mathcal{P}, r)$, with state space $\mathcal{S}$, action space $\mathcal{A}$, horizon $H$, transition function $\mathcal{P} : \mathcal{S} \times \mathcal{A} \to \Delta(\mathcal{S})$ and reward function $r : \mathcal{S} \times \mathcal{A} \to [0, 1]$. Without loss of generality, we assume a fixed start state $s_0$.

A policy $\pi : \mathcal{S} \to \Delta(\mathcal{A})$ specifies a decision-making strategy in which the agent chooses actions adaptively based on the current state, that is, $a \sim \pi(\cdot \mid s)$. The value function $V^\pi : \mathcal{S} \to \mathbb{R}$ and the action-value function (Q-function) $Q^\pi : \mathcal{S} \times \mathcal{A} \to \mathbb{R}$ are defined as

$$V^\pi(s) = \mathbb{E}_\pi \Big[ \sum_{t=1}^\infty r(s_t, a_t) \,\Big|\, s_0 = s \Big], \quad Q^\pi(s, a) = \mathbb{E}_\pi \Big[ \sum_{t=1}^\infty r(s_t, a_t) \,\Big|\, s_0 = s, a_0 = a \Big], \quad (1)$$

where the expectation is w.r.t. the trajectory $\tau$ induced by $\pi$. We define the Bellman evaluation operator as

$$(\mathbb{T}^\pi f)(s, a) = \mathbb{E}_{s' \sim \mathcal{P}(\cdot | s, a), a' \sim \pi(\cdot | s')} \big[ r(s, a) + \gamma f(s', a') \big]. \quad (2)$$

We use $\pi^\star$, $Q^\star$, and $V^\star$ to denote an optimal policy, the corresponding optimal Q-function and optimal value function, respectively. We have the Bellman optimality equation

$$V^\star(s) = \max_{a \in \mathcal{A}} Q^\star(s, a), \quad Q^\star(s, a) = \mathbb{E}_{s' \sim \mathcal{P}(\cdot | s, a)} \big[ r(s, a) + \gamma V^\star(s') \big]. \quad (3)$$

Meanwhile, the optimal policy $\pi^\star$ satisfies $\pi^\star(\cdot \mid s) = \operatorname{argmax}_\pi \langle Q^\star(s, \cdot), \pi(\cdot \mid s) \rangle_{\mathcal{A}}$. We aim to learn a policy $\pi$ from the candidate policy class $\Pi$ that maximizes the expected cumulative reward. Correspondingly, we define the performance metric as the sub-optimality compared with the optimal policy, i.e.,

$$\text{SubOpt}(\pi) = V^{\pi^\star}(s_0) - V^\pi(s_0). \quad (4)$$

### 2.1 BELLMAN CONSISTENT PESSIMISM

A unique challenge in offline RL is that the learned policy may induce a state-action density that is different from the data distribution $\mu$, which may lead to large extrapolation errors when we do not impose any coverage assumption on $\mu$. Therefore, it is important to carefully characterize the distribution shift, which we measure using the coverage coefficient. Specifically, we adopt the one used in Xie et al. (2021) that considers the distribution shift of Bellman errors:

**Definition 1** (Bellman shift coefficient (Xie et al., 2021)). *We define $\mathcal{C}(\nu; \mu, \mathcal{Q}, \pi)$ as follows to measure the distribution shift from an arbitrary distribution $\nu$ to the data distribution $\mu$, w.r.t. $\mathcal{Q}$ and $\pi$,*

$$\mathcal{C}(\nu; \mu, \mathcal{Q}, \pi) := \max_{q \in \mathcal{Q}} \frac{\|q - \mathbb{T}^\pi q\|_{2,\nu}^2}{\|q - \mathbb{T}^\pi q\|_{2,\mu}^2}.$$

Here $\mathcal{Q}$ is the Q-function approximation class we consider. Intuitively, $\mathcal{C}(\nu; \mu, \mathcal{Q}, \pi)$ measures how well Bellman errors under $\pi$ transfer between the distributions $\nu$ and $\mu$. For instance, a small value of $\mathcal{C}(d^\pi; \mu, \mathcal{Q}, \pi)$ enables accurate policy evaluation for $\pi$ using data collected under $\mu$. Definition 1 is a generalization compared to prior works that is defined specific to linear function approximation (Agarwal et al., 2021; Jin et al., 2021). More generally, we have $\mathcal{C}(\nu; \mu, \mathcal{Q}, \pi) \leq \|\nu/\mu\|_\infty := \sup_{s,a} \frac{\nu(s,a)}{\mu(s,a)}$ holds for any $\pi$ and $\mathcal{Q}$.

## 2.2 PREFERENCE-BASED REINFORCEMENT LEARNING

To learn reward functions from preference labels, we consider the Bradley-Terry pairwise preference model (Bradley & Terry, 1952) as used by most prior works (Christiano et al., 2017; Ibarz et al., 2018; Palan et al., 2019). Specifically, the preference label between two given trajectories $\tau_i$ and $\tau_j$ is defined as

$$\mathbb{P}\left(\tau_i \succ \tau_j \mid R\right) = \frac{1}{\exp\left(R(\tau_j) - R(\tau_i)\right) + 1}, \tag{5}$$

where $\tau = (s_t, a_t)_{t=0}^T$ is a trajectory and $R(\tau) = \sum_{t=0}^T \gamma^t r(s_t, a_t)$ is the return function. To simplify the theoretical analysis, we consider learning a *return* model instead of a *reward* model. The return model $\widehat{R}$ is trained to minimize the cross-entropy loss between the predicted preference and the ground truth with a given preference dataset $\mathcal{D}_{\text{pref}}$ as follows:

$$\mathcal{L}_{\text{CE}}(R) = - \mathbb{E}_{(\tau^1, \tau^2, o) \sim \mathcal{D}_{\text{pref}}} \left[ o \log \mathbb{P}\left(\tau_1 \succ \tau_2 \mid R\right) + (1-o) \log(1 - \mathbb{P}\left(\tau_1 \succ \tau_2 \mid R\right)) \right], \tag{6}$$

where $o$ is the ground truth label given by human labelers. We assume that the difference of return functions $\Delta\mathcal{R} := \{\Delta R(\tau_1, \tau_2) : \text{Traj} \times \text{Traj} \to \mathbb{R} | \exists R \in \mathcal{R}, \Delta R(\tau_1, \tau_2) = R(\tau_1) - R(\tau_2)\}$ has a finite Eluder dimension, which is a common general function approximation class in RL literature (Russo & Van Roy, 2013; Chen et al., 2022).

**Definition 2** (Eluder Dimension (Russo & Van Roy, 2013)). *Suppose $\mathcal{F}$ is a function class defined in $\mathcal{X}$, the $\alpha$-Eluder dimension $d_{Elu}(\mathcal{F}, \alpha)$ is the longest sequence $\{x_1, x_2, \cdots, x_n\} \in \mathcal{X}$ such that there exists $\alpha' \geq \alpha$ where $x_i$ is $\alpha'$-independent of $\{x_1, \cdots, x_{i-1}\}$ for all $i \in [n]$.*

The above defined Eluder dimension is used to establish a suboptimality upper bound for our proposed algorithm in Section 4. The following generalized linear preference model considered by many prior works (Pacchiano et al., 2021; Zhan et al., 2023b) is a special case of finite Eluder dimension (Chen et al., 2022). We include it to show that our analysis in Section 4 is general.

**Definition 3** (Generalized Linear Preference Model). *In $d$-dimensional generalized linear models, the preference function can be represented as $\mathbb{P}(\tau_1 \succ \tau_2 | \theta) = \sigma(\langle \phi(\tau_1, \tau_2), R \rangle)$ where $\sigma$ is an increasing Lipschitz continuous function, $\phi : \text{Traj} \times \text{Traj} \to \mathbb{R}^d$ is a known feature map satisfying $\|\phi(\tau_1, \tau_2)\|_2 \leq H$ and $\theta \in \mathbb{R}^d$ is the unknown parameter.*

## 3 METHOD

In this section, we present our proposed algorithm, Offline Preference-based Reinforcement Learning with In-Dataset Exploration (OPRIDE), illustrated in Figure 1. The key idea of OPRIDE is to enhance the query efficiency of offline PbRL by conducting optimistic exploration with in-dataset queries and then utilizing the learned reward function pessimistically with discount factor scheduling. The overall algorithm is shown in Algorithm 1.

## 3.1 Offline Query Selection with In-Dataset Exploration

Generating informative queries is crucial for calibrating the reward function. Various methods have been proposed to generate queries for offline preference-based RL, like disagreement-based approaches (Christiano et al., 2017) and information-gain-based approaches (Wilson et al., 2012; Shin et al., 2023), but it may lead the algorithm to focus on refining reward estimates in regions of the state space that are irrelevant to the optimal policy. This naturally leads to the idea of employing an exploration objective (Akrour et al., 2011) into offline query selection, where we maximize the information gain about the *optimal policy* rather than the *reward function*.

Inspired by principled exploration strategies for PbRL, analyzed in Section 4, we propose to use the difference of value differences as the exploration criteria. Specifically, we first train a set of reward functions $\{r_{\theta_i}\}_{i=1}^M$ using bootstapping, then train a set of value functions $\{V_{\psi_i}\}_{i=1}^M$ using offline algorithms like IQL (Kostrikov et al., 2021; Ma et al., 2021) with the reward functions. Finally, we select two trajectories $\tau_1$ and $\tau_2$ that maximize the difference of value differences between the two trajectories:

$$\underset{(\tau_1,\tau_2)\in\mathcal{D}}{\operatorname{argmax}} \underset{i,j\in[M]}{\operatorname{argmax}} \left|\left(V_{\psi_i}(\tau_1) - V_{\psi_j}(\tau_1)\right) - \left(V_{\psi_i}(\tau_2) - V_{\psi_j}(\tau_2)\right)\right|, \tag{7}$$

The reward function $r_{\theta_i}$ and the value functions $Q_{\phi_i}, V_{\psi_i}$ are iteratively updated after each preference query. Intuitively, Equation 7 aims to select queries that most effectively minimize the diameter of the uncertainty set for the value function. The diameter represents the maximum possible disagreement between any two candidate value functions on any two policies. Reducing this diameter is proportional to the information gain from a query. Therefore, minimizing this diameter is equivalent to maximizing the information gain for each query. This objective is directly linked to the information gain via information ratio $\Gamma$ (Lu & Van Roy, 2019; Russo & Van Roy, 2016). The diameter of the uncertainty set is upper-bounded by $P\left(\operatorname{diam}(\mathcal{R}) \leq \Gamma_\delta \sqrt{I(\mathcal{R};\mathcal{D})}\right) \geq 1 - \delta$, where $\operatorname{diam}(\mathcal{R}) = \max_{R_1,R_2\in\mathcal{R}} \max_{\pi_1,\pi_2\in\Pi} |(R_1(\pi_1) - R_1(\pi_2)) - (R_2(\pi_1) - R_2(\pi_2))|$, and $I(\mathcal{R};\mathcal{D})$ is the mutual information between the reward function class $\mathcal{R}$ and the query dataset $\mathcal{D}$. Maximizing the information gain $I$ directly corresponds to reducing the diameter. Mathematically, the sample complexity of reward function estimation is proportional to the Eluder dimension of the reward function class, $d_{\mathrm{Elu}}(\mathcal{R})$, while Equation 7's complexity relates to the Eluder dimension of the optimal value function class. It is often the case that $d_{\mathrm{Elu}}(\mathcal{V}^*) \ll d_{\mathrm{Elu}}(\mathcal{R})$. Please refer to Section 4 for the complete theoretical analysis.

## 3.2 Policy Extraction with Variance-based Discount Scheduling

After obtaining the preference feedback, we can train the reward function using the cross-entropy loss in Equation 6 and annotate the reward-free dataset $\mathcal{D} = \{\{(s_t^n, a_t^n)\}_{t=0}^T\}_{n=1}^N$ to obtain a labeled dataset $\widehat{\mathcal{D}} = \{\{(s_t^n, a_t^n, \widehat{r}_t^n)\}_{t=0}^T\}_{n=1}^N$ where $\widehat{r} = 1/M \sum_{i=1}^M r_{\theta_i}$. However, it is well-known that a learned reward function is prone to overoptimization (Gao et al., 2023; Zhu et al., 2024), leading to overestimation of the value function and, subsequently, a suboptimal policy.

Learning from preference feedback is more vulnerable to this issue, as the feedback is binary and sparse. To solve this issue, we propose to adjust the discount factor based on the variance of the value function estimates that serve as a stronger regulator. Using a smaller discount factor is known to provide pessimistic and robust guarantees and performs well in various settings like imitation learning (Liu et al., 2024). Specifically, we reduce the discount factor where there is a higher variance in value estimation, thereby alleviating the impact of reward function overestimation.

$$\widehat{\gamma}(s, a) = \begin{cases} \gamma_{\mathrm{small}}, & \text{if} \quad \mathrm{Var}\{Q_{\phi_i}(s,a)\}_{i=1}^M > \mathrm{Top}\ m\%(\{\mathrm{Var}_j\{Q_{\phi_i}(s_j,a_j)\}_{i=1}^M\}_{j=1}^{|\mathrm{Batch}|}) \\ \gamma, & \text{else} \end{cases} \tag{8}$$

where $\widehat{\gamma}$ is the adjusted discount factor. Please note that if the variance of the value estimation for a data point is greater than the top $m\%$ in the batch, we consider that the reward function for this data point has overestimation noise and reduces the corresponding discount factor. Subsequently, we can learn a corresponding Q-value function and extract the policy from the labeled datasets $\widehat{\mathcal{D}}$ by adopting the standard offline reinforcement learning algorithms, like IQL (Kostrikov et al., 2021). We also consider a softer confidence discount mechanism, which is detailed in the Appendix E.

## 4 THEORETICAL ANALYSIS

In this section, we investigate the theoretical guarantees for generating queries with an explorative objective. Specifically, we consider the strategy consisting of the following procedures: (1) construct a confidence set for the return function based on existing queries; (2) construct a candidate policy set using pessimistic value estimation as the criteria; and (3) select a pair of policies that maximize disagreement on values for new queries. A detailed strategy description is available in Algorithm 2.

**Construct Confidence Set.** For the return function, we can use the cross entropy loss as in Equation 6 to get the maximum likelihood estimator (MLE) for the return function $\widehat{R}_k$:

$$\widehat{R}_k = \underset{R \in \mathcal{R}}{\operatorname{argmin}} \, L_k(R), \tag{9}$$

where $L_k(R) = \sum_{i=1}^{k} (o_i \log \mathbb{P}(\tau_i^1 \succ \tau_i^2; R) + (1 - o_i) \log(1 - \mathbb{P}(\tau_i^1 \succ \tau_i^2; R)))$ is the MLE loss. Then we can constuct the confidence set for the reward function as follows:

$$\mathcal{C}_k(\mathcal{R}) = \left\{ R \in \mathcal{R} \,\Big|\, \sum_{i=1}^{k} \left( (R(\tau_i^1) - R(\tau_i^2)) - (\widehat{R}_k(\tau_i^1) - \widehat{R}_k(\tau_i^2)) \right)^2 \le \beta_k \right\} \tag{10}$$

where $\beta_k$ is the confidence parameter to be specified later. Given the confidence set for the return function $R$, we can subsequently construct a confidence set for policies using a pessimistic value function. Specifically, we consider the pessimistic value function $\widehat{q}_R$ that leads to the worst-case value for the optimal policy over the Bellman uncertainty of the value function. Please refer to Algorithm 3 in Appendix A.1 for more details. The candidate policy set $\Pi_k$ is constructed as follows:

$$\Pi_k = \left\{ \widehat{\pi} \,\Big|\, \exists \, R \in \mathcal{C}_k(\mathcal{R}), \widehat{\pi} = \underset{\pi \in \Pi}{\operatorname{argmax}} \, \widehat{q}_R(s_1, \pi). \right\}. \tag{11}$$

Intuitively speaking, $\Pi_k$ consists of policies that are possibly optimal within the current level of uncertainty over reward and dynamics. By constraining exploration policies in $\Pi_k$, we achieve proper exploitation by avoiding unnecessary explorations.

**Selecting Exploratory Policies.** For a given pair of policies $(\pi_1, \pi_2)$ in $\Pi_k$, we determine their exploration power by measuring how much disagreement can be made for different reward functions in the confidence set. Specifically, we select explorative policies via the following criteria:

$$\pi_1^k, \pi_2^k = \underset{\pi_1, \pi_2 \in \Pi_k}{\operatorname{argmax}} \, \max_{R_1, R_2 \in \mathcal{C}_k(\mathcal{R})} \left( \left( \widehat{v}_{R_1}^{\pi_1} - \widehat{v}_{R_2}^{\pi_1} \right) - \left( \widehat{v}_{R_1}^{\pi_2} - \widehat{v}_{R_2}^{\pi_2} \right) \right). \tag{12}$$

Intuitively, we choose two policies $\pi_1, \pi_2$ such that there is a $R_1 \in \mathcal{C}_k(\mathcal{R})$ that strongly prefers $\pi_1$ over $\pi_2$, and there is a $R_2 \in \mathcal{C}_k(\mathcal{R})$ that strongly prefer $\pi_2$ over $\pi_1$. We sample two trajectories $\tau^{k,1} \sim \pi^{k,1}, \tau^{k,2} \sim \pi^{k,2}$, query the preference between them, and add it to the preference dataset. Choosing the pair of trajectories that maximize disagreement helps us explore efficiently. Then, we have the following theoretical guarantee:

**Theorem 4.** *Let $\beta_k = c_1 \sqrt{\log(K|\Delta\mathcal{R}|)/K}$ and $\epsilon = c_2 \sqrt{\log(N|\Pi||\mathcal{Q}|)/N}$, where $c_1, c_2$ are universal constants. Then the expected suboptimality of $\bar{\pi}$ from Algorithm 2 is upper bounded by*

$$\operatorname{SubOpt}(\bar{\pi}) \le \mathcal{O}\left( \underbrace{\sqrt{\frac{C^\dagger \log(N|\mathcal{Q}||\Pi|)}{N(1-\gamma)^2}}}_{\textit{Offline Error}} + \underbrace{\sqrt{\frac{\kappa d_{Elu}(\Delta\mathcal{R}, 1/K) \log(K|\Delta\mathcal{R}|)}{K(1-\gamma)}}}_{\textit{Preference Error}} \right), \tag{13}$$

*where $\kappa$ is the degree of non-linearity of the link function $\sigma$, $C^\dagger$ is the coverage coefficient in Definition 1, $N$ is the size of the offline dataset and $K$ is the number of queries.*

*Proof.* See Appendix B for a detailed proof. □

---

**Algorithm 1** Offline Preference-Based Reinforcement Learning with In-Dataset Exploration

---

1: **Input**: Unlabeled offline dataset $\mathcal{D} = \{\tau_n = \{(s_t^n, a_t^n)\}_{t=0}^T\}_{n=1}^N$, query budget $K$, ensemble number $M$
2: Initialized the preference dataset $\mathcal{D}_{\text{pref}} \leftarrow \varnothing$.
3: **for** episode $k = 1, \cdots, K$ **do**
4:      Train $M$ ensembles of reward network $r_{\theta_i}$ with $\mathcal{D}_{\text{pref}}$ using $\mathcal{L}_{\text{CE}}$ in Equation 6.
5:      Train $M$ corresponding value functions $V_{\psi_i}, Q_{\phi_i}$ with each reward function $r_{\theta_i}$.
6:      Select trajectories $\tau^{k,1}, \tau^{k,2}$ that maximize the exploration objective according to Equation 7.
7:      Receive the preference $o_k$ between $\tau^{k,1}$ and $\tau^{k,2}$ and add it to the preference dataset, i.e.,

$$\mathcal{D}_{\text{pref}} \leftarrow \mathcal{D}_{\text{pref}} \cup \{(\tau^{k,1}, \tau^{k,2}, o_k)\}.$$

8: **end for**
9: Annotate the unlabeled offline dataset $\mathcal{D}$ with the reward function $\widehat{\theta}$ and obtain $\widehat{\mathcal{D}}$.
10: Adjust the discount facto $\gamma$ to $\widehat{\gamma}$ based on Equation 8.
11: Extract policy $\pi_\xi$ via offline RL from $\widehat{\mathcal{D}}$.
12: **Output**: The learned policy $\pi_\xi$

---

| Domain | Task | OPRL | PT | PT+PDS | IDRL | OPRIDE |
|--------|------|------|-----|--------|------|--------|
| | lever-pull | **63.2±10.4** | 49.2±3.7 | 51.7±0.1 | 33.1±1.2 | 51.8±1.6 |
| | peg-insert-side | 3.5±1.8 | 16.8±0.1 | 12.4±1.4 | 67.4±0.1 | **79.0±0.2** |
| | plate-slide | 77.4±1.6 | 4.9±0.0 | 37.3±2.3 | **79.6±3.5** | **79.9±4.6** |
| | push | 10.6±1.5 | 16.7±5.0 | 1.8±0.4 | 30.7±5.3 | **59.1±5.4** |
| | push-back | 0.8±0.0 | 1.1±0.4 | 1.1±0.1 | 14.0±1.1 | **17.7±2.0** |
| Metaworld | push-wall | 7.4±4.2 | 74.8±14.4 | 3.4±0.9 | 89.2±3.2 | **102.2±1.2** |
| | reach | 63.5±2.9 | 82.0±0.8 | 84.3±0.9 | 75.8±1.8 | **88.0±0.5** |
| | soccer | 34.3±4.0 | **51.3±4.1** | 41.5±11.9 | 44.3±2.1 | 45.4±3.9 |
| | sweep-into | 37.1±13.9 | 9.8±0.2 | 9.2±0.1 | 63.1±3.5 | **71.6±0.1** |
| | sweep | 6.8±1.8 | 8.0±0.4 | 8.0±0.1 | 73.0±2.8 | **78.5±1.0** |
| Average | | 30.4±4.2 | 31.4±2.9 | 25.0±1.8 | 57.0±6.3 | **65.3±3.3** |

Table 1: Performance of offline RL algorithm on the reward-labeled dataset with different preference reward learning methods on the Meta-World tasks over five random seeds.

Equation 13 decomposes the suboptimality of Algorithm 2 into two terms nicely: the offline error term and the preference error term. The first error is due to the finite sample bias of the dataset, and the preference error is due to the limited amount of preference queries. Compared to pure online learning, the preference error is reduced by a factor of $1/(1-\gamma)$. Therefore, querying with an offline dataset can be much more sample-efficient than pure online queries when $N \gg K$. This is because the offline dataset contains rich information about dynamics and can reduce the effective horizon of the problem (Hu et al., 2023). This also aligns with our empirical findings that $\sim 10$ queries are usually sufficient for reasonable performance in offline settings.

## 5 EXPERIMENTS

In this section, we aim to answer the following questions: (1) How does our method perform on various navigation and manipulation tasks compared to other offline PbRL methods? (2) How effective is the proposed exploration-based query selection and discounted-based pessimism? (3) How does our method perform across different numbers of queries?

### 5.1 EXPERIMENTAL DETAILS

**Environment Setup.** We perform empirical evaluations on Meta-World (Yu et al., 2019) and the Antmaze task on the D4RL benchmark (Fu et al., 2020). In the preference query, we use a segment length of 50 for all tasks. Our setup begins with a dataset of unlabeled trajectories. We use a

| Domain | Task | OPRL | PT | PT+PDS | IDRL | OPRIDE |
|--------|------|------|-----|--------|------|--------|
| Antmaze | umaze | 76.3±3.7 | 77.5±4.5 | 84.5±8.5 | 85.5±3.4 | **87.5±5.6** |
| | umaze-diverse | 72.5±3.4 | 68.0±3.0 | **78.0±6.0** | 69.1±4.2 | 73.1±2.4 |
| | medium-play | 0.0±0.0 | 63.5±0.5 | **72.5±6.5** | 63.8±4.1 | 62.2±2.0 |
| | medium-diverse | 0.0±0.0 | 63.5±4.5 | 58.0±4.0 | 65.7±4.1 | **69.4±5.2** |
| | large-play | 7.3±0.9 | 6.5±2.5 | 9.0±8.0 | 18.7±3.4 | **27.5±12.5** |
| | large-diverse | 6.9±2.4 | **23.5±0.5** | 8.5±2.5 | 14.3±2.5 | 21.5±1.5 |
| Average | | 27.1±1.7 | 50.4±2.5 | 51.7±5.9 | 52.8±3.6 | **56.8±4.8** |

Table 2: Performance of offline RL algorithm on the reward-labeled dataset with different preference reward learning methods on the Antmaze tasks over five random seeds.

| Task | PT | PDS + Random Query | VDS + Random Query | VDS + Disagreement | OPRIDE (VDS+IDE) |
|------|-----|--------------------|--------------------|--------------------|--------------------|
| bin-picking | 31.9±16.2 | 53.4±19.0 | 71.9±9.0 | 78.5±17.8 | **93.3±3.2** |
| button-press-wall | 58.8±0.9 | 59.4±0.9 | **77.2±0.8** | 67.4±5.4 | **77.7±0.1** |
| door-close | 65.1±10.1 | 62.4±8.7 | 72.3±0.1 | 88.3±0.7 | **94.8±1.1** |
| faucet-close | 57.8±0.9 | 46.2±0.2 | 59.4±8.5 | 48.7±0.6 | **73.1±0.8** |
| peg-insert-side | 16.8±0.1 | 12.4±1.4 | 13.8±4.4 | 9.7±8.5 | **79.0±0.2** |
| reach | 82.0±0.8 | 84.3±0.9 | 83.3±0.1 | **86.6±0.1** | **88.0±0.5** |
| sweep | 8.0±0.4 | 8.0±0.1 | 28.7±1.8 | 18.2±2.9 | **78.5±1.0** |

Table 3: Ablation of the query selection module on the Meta-World tasks. We report the performance of offline RL algorithm on the reward-labeled dataset with various query selection and policy extration mechanism. IDE and VDS represent the In-Dataset Exploration module and the Variance-based Discount Scheduling module proposed in Section 3.1, respectively.

preference-based RL method (e.g., OPRIDE) to select queries from this dataset. These queries are then used to train a reward model. Subsequently, this learned reward model is used to relabel the entire trajectory dataset with rewards, which is then used to train a policy with a standard offline RL algorithm. We adopt the normalized score metric proposed by the D4RL benchmark. Please refer to Appendix F for more experimental details.

**Baselines.** We compare OPRIDE with several state-of-the-art offline PbRL methods, including Offline Preference-based Reinforcement Learning (OPRL; Shin et al., 2023), Preference Transformer (PT; Kim et al., 2023) and Information Directed Reward Learning (IDRL; Lindner et al., 2021). To illustrate the effectiveness of our proposed variance-based discount, we compare our method with Provable Data Sharing (PDS; Hu et al., 2023) as a baseline algorithm. For OPRL, we employ disagreement-based query selection, as it yields the best performance. We adopt the same architecture as in Preference Transformer (PT) for a fair comparison.

## 5.2 EXPERIMENTAL RESULTS

**Answer to Question 1:** To show that OPRIDE can generate valuable rewards with a few queries, we conducted a comprehensive comparative analysis of OPRIDE against several baseline methods, utilizing Meta-World and Antmaze tasks as our testing grounds. Specifically, we use a budget of 10 queries on each task for all offline preference-based reinforcement learning methods. Then, we let all algorithms employ the IQL algorithm for subsequent offline training for a fair comparison. The experimental results in Table 1 and Table 2 are normalized episode returns averaged over five random seeds. In most tasks in Meta-World and Antmaze, OPRIDE demonstrates superior performance compared to baseline algorithms. Moreover, unlike IDRL, which relies on the Laplacian approximation and the Hessian matrix for posterior computation, our method leverages critic values for query selection, ensuring easier implementation and superior empirical performance, as demonstrated in our comparative experiments.

| Domain | Tasks | Zero | Random | Negative | OPRIDE |
|--------|-------|------|--------|----------|--------|
| Metaworld | coffee-push | 7.6±4.3 | 5.8±2.7 | 0.7±0.1 | **59.4±24.8** |
| | disassemble | 9.3±0.4 | 16.8±7.3 | 10.1±0.2 | **26.6±4.9** |
| | hammer | 38.1±6.4 | 46.1±2.4 | 22.6±1.8 | **50.3±3.2** |
| | push | **57.5±1.5** | 34.4±17.3 | 4.6±2.3 | 59.1±5.4 |
| | push-wall | 81.9±3.8 | 80.1±0.9 | 17.6±1.9 | **102.2±1.2** |
| | soccer | 33.3±1.6 | 41.1±8.8 | 44.0±6.4 | **45.4±3.9** |
| | sweep | 29.0±0.2 | 29.0±2.6 | 24.9±0.3 | **78.5±1.0** |

Table 4: Comparison between the survival instinct and OPRIDE.

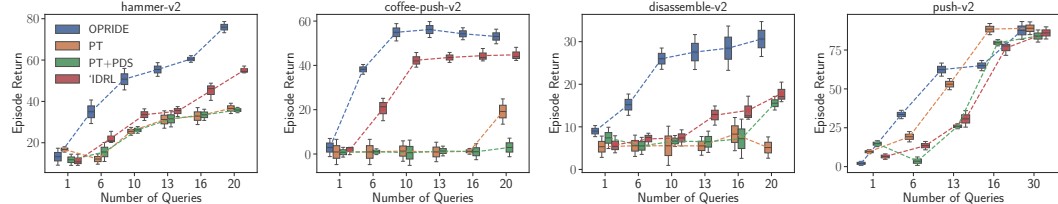

Figure 2: Performance of offline preference-based RL algorithms with various queries. OPRIDE achieves a better query efficiency across tasks and number of queries.

We also compare OPRIDE with the recent research work Survival Instinct (Li et al., 2024) since they find that wrong rewards can also lead to good offline RL performance. Specifically, we used three types of rewards, the same as the author: (1) zero: the zero reward, (2) random: labeling each transition with a reward value randomly sampled from Unif [0, 1], and (3) negative: the negation of true reward. Then, we trained the same offline learning algorithm as OPRIDE on the reward-labeled dataset. The experimental results in Table 4 indicate that OPRIDE still outperforms these baselines in most tasks. We attribute the above experimental results to the challenging nature of the dataset we created. Specifically, in Li et al. (2024), the perturbed script policy data accounts for 100% of the dataset. However, in our created dataset, the perturbed script policy data only accounts for 5% of the dataset. We conduct additional experiments on Mujoco and Kitchen tasks. Please refer to Appendix E for the complete experimental results.

**Answer to Question 2:** To study the contribution of each component in our framework, we conduct several ablation studies to verify the effectiveness of each part, as shown in Table 3. Comparing our method with the `VDS + Random Query` and the `VDS + Disagreement` baseline, we can see that disagreement-based approaches offer little improvement over the random query selection baseline, while our exploration criteria lead to vast performance improvement, showcasing that our method is able to collect useful information within a few queries. Comparing the `PDS + Random Query` and the `VDS + Random Query` baseline, we can conclude that while PDS is helpful on some tasks like `bin-picking-v2`, it fails to prevent reward overoptimization and makes the performance worse on some other tasks. On the contrary, `VDS + Random Query` is able to improve over the `PT` baseline on most tasks, showing its robust ability to reduce reward overestimation.

We have conducted ablation experiments to determine the sensitivity of the discount factor hyperparameter. Specifically, we vary the $\gamma_{small}$ values from 0.5 to 0.95 for the data points with the high variance. The experimental results in Table 7 in Appendix E indicate that 0.7∼0.8 is an appropriate range for $\gamma_{small}$, and the performance is robust across different $\gamma_{small}$ values. We conduct additional ablation studies for the In-Dataset Exploration module, the Variance-based Discount Scheduling module and the hyperparameter $m$. Please refer to Appendix E for the complete results.

**Answer to Question 3:** To investigate how the number of queries affects OPRIDE 's overall performance, we vary the number of queries and compare our method with various baselines. The results presented in Figure 2 demonstrate that OPRIDE achieves a superior query efficiency and significantly outperforms the baselines across various numbers of queries. In most tasks, OPRIDE achieves good performance with just ten queries, and its performance continues to improve as the number of queries increases. In contrast, the baseline methods require multiple times the number of queries to

achieve performance on par with OPRIDE (e.g., `hammer-v2`). Even with 20 queries, the baseline algorithm shows no significant improvement on some hard tasks (e.g., `coffee-push-v2`).

**Computational Cost:** To improve computational efficiency, we implement the following modifications. (1) When training reward functions, instead of traversing the entire dataset, we sample $S$ segments at each iteration. (2) Furthermore, we utilize a multi-head mechanism instead of separate training for each reward function. This means different reward functions share the same backbone, with only the last layer being distinct. Therefore, the computational cost of our overall method is $S \times (1 + (M-1)/L) \times K$ instead of $N \times M \times K$, where $L$ is the number of neural network layers. Please refer to Appendix E for the complete computational cost results.

## 6 CONCLUSION

This paper proposes a new framework, in-dataset exploration, to improve query efficiency in offline PbRL. Compared with disagreement-based approaches, using an exploration strategy helps reduce the burden of learning an accurate reward function in the low-return region, improving learning efficiency. Our proposed algorithm, OPRIDE, conducts in-dataset exploration by weighted trajectory queries, and a principled exploration strategy deals with pairwise queries. Our method has provable guarantees, and our practical variant achieves strong empirical performance on various tasks. Compared to prior methods, our method significantly reduces the required queries. Overall, our method provides a promising and principled way to reduce queries required from human labelers in PbRL.

## REPRODUCIBILITY STATEMENT

We have provided the source code in the supplementary materials, which will be made public after the paper is accepted. We have provided theoretical analysis in the Appendix A and Appendix B. We have also provided implementation details in the Appendix F.

## ACKNOWLEDGMENTS

This work is supported by the National Key R&D Program of China (No.2022ZD0116405) and Amazon Research Award.

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

# A  ADDITIONAL DETAILS

In this section, we provide a detailed description for the theoretical version of OPRIDE as in Algorithm 2.

---

**Algorithm 2** Theoretical Analysis Version of OPRIDE

1: **Input**: Unlabeled offline dataset $\mathcal{D}$, query budget $K$
2: Initialized preference dataset $\mathcal{D}_{\text{pref}} \leftarrow \emptyset$.
3: **for** $k = 1, \cdots, K$ **do**
4:    Calculate confidence set $\mathcal{C}_k(\mathcal{R})$ for reward function based on $\mathcal{D}_{\text{pref}}$ with Equation 10.
5:    Calculate pessimistic value function $\widehat{q}(\cdot)$ using Algorithm 3 for each reward function in $\mathcal{C}_k(\mathcal{R})$.
6:    Construct the near-optimal policy set $\Pi_k$ using Equation 11.
7:    Select explorative policies $\pi_k^1, \pi_k^2$ within $\Pi_k$ based on Equation 12.
8:    Sample trajectories $\tau_k^1, \tau_k^2$ with selected policy $\pi_k^1, \pi_k^2$.
9:    Receive the preference $o_k$ between $\tau_k^1$ and $\tau_k^2$ and add it to the preference dataset

$$\mathcal{D}_{\text{pref}} \leftarrow \mathcal{D}_{\text{pref}} \cup \{(\tau_k^1, \tau_k^2, o_k)\}.$$

10: **end for**
11: **Output**: Average policy $\bar{\pi} = \frac{1}{2K} \cdot \sum_{k=1}^{K} (\pi_k^1 + \pi_k^2)$.

---

## A.1  DETAILS OF BELLMAN-CONSISTENT PESSIMISM (BCP; XIE ET AL., 2021)

In this section, we consider *Bellman-consistent Pessimism* (BCP; Xie et al., 2021) as the backbone algorithm, described in Algorithm 3. It is a representative model-free offline algorithm with theoretical guarantees. PEVI uses negative bonus $\Gamma(\cdot, \cdot)$ over standard $Q$-value estimation $\widehat{Q}(\cdot, \cdot) = (\widehat{\mathbb{B}}\widehat{V})(\cdot)$ to reduce potential bias due to finite data, where $\widehat{\mathbb{B}}$ is some empirical estimation of $\mathbb{B}$ from dataset $\mathcal{D}$. We use the following notion of $\xi$-uncertainty quantifier as follows to formalize the idea of pessimism.

---

**Algorithm 3** Bellman-consistent Pessimism (BCP)

1: **Input**: Offline Dataset $\mathcal{D}_{\text{off}} = \{\tau_k = \{(s_t^k, a_t^k)\}_{t=0}^T\}_{k=1}^K$, reward function $r$.
2: Set the loss function as

$$\mathcal{L}(q, q', \pi; \mathcal{D}) = \sum_{k=1}^{K} \sum_{t=0}^{T} \left(q_t(s_t^k, a_t^k) - (r(s_t^k, a_t^k) + \gamma q'_{t+1}(s_{t+1}^k, \pi_{t+1}))\right)^2. \tag{14}$$

3: Set the confidence set of value functions as

$$\mathcal{V}(\pi, \epsilon) = \left\{ q \in \mathcal{V} : \mathcal{L}(q, q, \pi; \mathcal{D}) - \min_{q' \in \mathcal{V}} \mathcal{L}(q', q, \pi; \mathcal{D}) \leq \epsilon \right\}. \tag{15}$$

4: Compute pessimistic policy and value function as

$$\widehat{\pi} = \operatorname*{argmax}_{\pi \in \Pi} \min_{q \in \mathcal{V}(\pi, \epsilon)} q_1(s_1, \pi). \tag{16}$$

and

$$\widehat{q} = \operatorname*{argmin}_{q \in \mathcal{V}(\widehat{\pi}, \epsilon)} q_1(s_1, \widehat{\pi}). \tag{17}$$

5: **Output**: $\widehat{\pi}$ and $\widehat{q}$.

---

**Lemma 5.** *Under conditions of Theorem, let $C^\dagger = \mathcal{C}(d_{\pi^\star}; \mu, \mathcal{Q}, \pi^\star)$, we have*

$$V^\star(\pi^\star) - V^\star(\widehat{\pi}) \leq O\left(\sqrt{\frac{C^\dagger \log \frac{|\mathcal{Q}||\Pi|}{\delta}}{N(1-\gamma)^2}}\right), \tag{18}$$

*where $\widehat{\pi}$ is the output of Algorithm 3 with dataset $\mathcal{D}_{off}$ and return function R. Similarly, we have*

$$V^{\star}(\pi) - \widehat{v}(\pi) \leq O\left(\sqrt{\frac{C^{\dagger} \log \frac{|\mathcal{Q}||\Pi|}{\delta}}{N(1-\gamma)^2}}\right), \tag{19}$$

*where $\widehat{v}$ is the output of Algorithm 4 with dataset $\mathcal{D}_{off}$, policy $\pi$ and return function R.*

*Proof.* This proof is mainly adapted from the proof of Theorem 1 in Xie et al. (2021) to the finite-horizon case. For simplicity we only prove the first part of the lemma. The second part can be proved similarly using the pessimistic property of the value function $\widehat{v}$.

Using the optimality of $\widehat{\pi}$, we have

$$\max_{v \in \mathcal{Q}_{\pi,\epsilon_r}} v(s_0, \pi) - \min_{v \in \mathcal{Q}_{\widehat{\pi},\epsilon_r}} v(s_0, \widehat{\pi}) \leq \max_{v \in \mathcal{Q}_{\pi,\epsilon_r}} v(s_0, \pi) - \min_{v \in \mathcal{Q}_{\pi,\epsilon_r}} v(s_0, \pi).$$

Now, let $v_{\min}(\pi) := \operatorname{argmin}_{v \in \mathcal{Q}_{\pi,\epsilon_r}} v(s_0, \pi)$ and $v_{\max}(\pi) := \operatorname{argmax}_{v \in \mathcal{Q}_{\pi,\epsilon_r}} v(s_0, \pi)$.

Using a standard reward decomposition argument Cai et al. (2020), we have

$$
\begin{aligned}
& v_{1,max}(\pi) - v_{1,min}(\pi) \\
&= v_{1,max} - v_1(\pi) + v_1(\pi) - v_{1,min} \\
&= \mathbb{E}_{d_\pi}\left[\sum_{h=1}^{H}(v_{h,max} - \mathbb{T}^\pi v_{h+1,max}) - \sum_{h=1}^{H}(v_{h,min} - \mathbb{T}^\pi v_{h+1,min})\right] \\
&\leq \sum_{h=1}^{H} \|v_{h,max} - \mathbb{T}^\pi v_{h+1,max}\|_{2,d^\pi} + \|v_{h,min} - \mathbb{T}^\pi v_{h+1,min}\|_{2,d^\pi} \\
&\leq \sqrt{\mathcal{C}(d^\pi; \mu, \mathcal{V}, \pi)} \sum_{i=1}^{H}(\|v_{h,max} - \mathbb{T}^\pi v_{h+1,max}\|_{2,\mu} + \|v_{h,min} - \mathbb{T}^\pi v_{h+1,min}\|_{2,\mu}) \\
&\leq \frac{1}{1-\gamma}\sqrt{\mathcal{C}(d^\pi; \mu, \mathcal{V}, \pi)\epsilon_b}, \tag{20}
\end{aligned}
$$

$$\tag{21}$$

holds under event $\mathcal{E}_2$ in Lemma 14 and $\mathcal{E}_3$ in Lemma 15. The second inequality follows from the definition of $\mathcal{C}(d^\pi; \mu, \mathcal{V}, \pi)$ and the last inequality follows from Lemma 14 and Lemma 15. Let $\pi = \pi^\star$ and plug in the definition of $\epsilon_b$, we complete the proof.

$\square$

---

**Algorithm 4** Bellman-consistent Pessimism Evaluation

---

1: **Input**: Offline Dataset $\mathcal{D}_{off} = \{\tau_k = \{(s_t^k, a_t^k)\}_{t=0}^{T}\}_{k=1}^{K}$, reward function $r$, policy $\pi$
2: Set the loss function as

$$\mathcal{L}(q, q', \pi; \mathcal{D}) = \sum_{k=1}^{K}\sum_{t=0}^{T}\left(q_t(s_t^k, a_h^k) - (r(s_t^k, a_t^k) + \gamma q'(s_{t+1}^k, \pi_{t+1})))\right)^2. \tag{22}$$

3: Set the confidence set of value functions as

$$\mathcal{V}(\pi, \epsilon) = \left\{q \in \mathcal{V} : \mathcal{L}(q, q, \pi; \mathcal{D}) - \min_{q' \in \mathcal{V}} \mathcal{L}(q', q, \pi; \mathcal{D}) \leq \epsilon\right\}. \tag{23}$$

4: Compute pessimistic value function as

$$\widehat{q} = \operatorname*{argmin}_{q \in \mathcal{V}(\pi,\epsilon)} q(s_0, \pi). \tag{24}$$

5: **Output**: $\widehat{v}$ and $\widehat{q}$.

---

# B Proof of Theorem 4

**Theorem 6** (Restatement of Theorem 4). *Suppose (1) $Q^\star \in \mathcal{Q}, \pi^\star \in \Pi$, and (2) $\mathbb{T}^\pi q \in \mathcal{Q}, \forall \pi \in \Pi, q \in \mathcal{Q}$. Also suppose the difference of return functions has a finite Eluder dimension $d_{Elu}(\Delta R, \alpha)$ and the underlying distribution of the offline dataset admit a finite coverage coefficient $C^\dagger$. Let $\beta_k = c_1\sqrt{\log(K|\Delta \mathcal{R}|)/K}$ and $\epsilon = c_2\sqrt{\log(N|\Pi||\mathcal{Q}|)/N}$, where $c_1, c_2$ are universal constants. Then the expected suboptimality of $\bar{\pi}$ from Algorithm 2 is upper bounded by*

$$
\text{SubOpt}(\bar{\pi}) \leq \mathcal{O}\left(\sqrt{\frac{C^\dagger \log(N|\mathcal{Q}||\Pi|)}{N(1-\gamma)^2}} + \sqrt{\frac{d_{Elu}(\Delta \mathcal{R}, 1/K)\log(K|\Delta \mathcal{R}|)}{K(1-\gamma)}}\right), \tag{25}
$$

*where $N$ is the size of the offline dataset and $K$ is the number of queries.*

**Remark 7.** *In Theorem 6 we consider finite function classes for policy $\Pi$, $Q$-value $\mathcal{Q}$ and return function $\mathcal{R}$. However, it can be readily extended to infinite function classes by using the covering number of the function classes, as done in Chen et al. (2022); Xie et al. (2021). We also remark that while we consider the realizable and Bellman-complete setting where $Q^\star \in \mathcal{Q}$ and $\mathbb{T}Q \in \mathcal{Q}$ for simplicity, we can extend the result to approximate realizable and Bellman-complete setting as in Xie et al. (2021).*

**Remark 8.** *The suboptimality bound uses the Eluder dimension of the difference function class $\Delta \mathcal{R}$ of the original return function class $\mathcal{R}$. This is because we can only determine $R(\tau_1) - R(\tau_2)$ from the preference query between $\tau_1$ and $\tau_2$ and the absolute value for $R(\tau)$ can be free to choose.*

*Proof.* For simplicity we let $V^\pi := V_1^\pi(s_1)$.

For any return function $\widetilde{R} \in \mathcal{C}_k(\mathcal{R})$ and the policy $\widetilde{\pi} = \text{BCP}(\mathcal{D}, \widetilde{R})$ generated by Algorithm 3, we have

$$
\begin{aligned}
&V_{R^\star}^{\pi^\star} - V_{R^\star}^{\widetilde{\pi}} \tag{26}\\
=&V_{R^\star}^{\pi^\star} - \widehat{v}_{R^\star}^{\pi^\star} + \widehat{v}_{R^\star}^{\pi^\star} - \widehat{v}_{\widetilde{R}}^{\pi^\star} + \widehat{v}_{\widetilde{R}}^{\pi^\star} - \widehat{v}_{\widetilde{R}}^{\widetilde{\pi}} + \widehat{v}_{\widetilde{R}}^{\widetilde{\pi}} - \widehat{v}_{R^\star}^{\widetilde{\pi}} + \widehat{v}_{R^\star}^{\widetilde{\pi}} - V_{R^\star}^{\widetilde{\pi}}\\
\leq&V_{R^\star}^{\pi^\star} - \widehat{v}_{R^\star}^{\pi^\star} + \widehat{v}_{R^\star}^{\pi^\star} - \widehat{v}_{\widetilde{R}}^{\pi^\star} + \widehat{v}_{\widetilde{R}}^{\pi^\star} - \widehat{v}_{\widetilde{R}}^{\widetilde{\pi}} + \widehat{v}_{\widetilde{R}}^{\widetilde{\pi}} - \widehat{v}_{R^\star}^{\widetilde{\pi}} + 0\\
\leq&V_{R^\star}^{\pi^\star} - \widehat{v}_{R^\star}^{\pi^\star} + \widehat{v}_{R^\star}^{\pi^\star} - \widehat{v}_{\widetilde{R}}^{\pi^\star} + 0 \qquad\qquad + \widehat{v}_{\widetilde{R}}^{\widetilde{\pi}} - \widehat{v}_{R^\star}^{\widetilde{\pi}}\\
\leq&V_{R^\star}^{\pi^\star} - \widehat{v}_{R^\star}^{\pi^\star} + \max_{R_1, R_2 \in \mathcal{C}_k(\mathcal{R})}\left(\widehat{v}_{R_1}^{\pi^\star} - \widehat{v}_{R_2}^{\pi^\star} + \widehat{v}_{R_2}^{\widetilde{\pi}} - \widehat{v}_{R_1}^{\widetilde{\pi}}\right)\\
\leq&V_{R^\star}^{\pi^\star} - \widehat{v}_{R^\star}^{\pi^\star} + \max_{R_1, R_2 \in \mathcal{C}_k(\mathcal{R})}\left(\widehat{v}_{R_1}^{\widetilde{\pi}^{k,1}} - \widehat{v}_{R_2}^{\widetilde{\pi}^{k,1}} + \widehat{v}_{R_2}^{\widetilde{\pi}^{k,2}} - \widehat{v}_{R_1}^{\widetilde{\pi}^{k,2}}\right), \tag{27}
\end{aligned}
$$

which hold under event $\mathcal{E}_1$ in Lemma 11. The first inequality follows from the pessimistic property of $\widehat{v}$, the second inequality follows from the fact that $\widetilde{\pi}$ is the optimal policy with respect to $\widehat{v}_{\widetilde{R}}$. The third inequality holds since $\widetilde{R}, R^\star \in \mathcal{C}_k(\mathcal{R})$ and the last inequality follows from the definition of $\widetilde{\pi}^{k,1}, \widetilde{\pi}^{k,2}$.

Following Lemma 5, we have for all policy $\pi$ and reward function $R$, the following holds with probability at least $1 - 2\delta$:

$$
|V_R^\pi - \widehat{v}_R^\pi| \leq c \cdot \sqrt{\frac{C^\dagger \log(N|\Pi||\mathcal{Q}|)}{N(1-\gamma)^2}} := \mathcal{E}_{\text{off}}.
$$

Then we have

$$V_{R^\star}^{\pi^\star} - V_{R^\star}^{\widetilde{\pi}} \tag{28}$$

$$\leq V_{R^\star}^{\pi^\star} - \widehat{v}_{R^\star}^{\pi^\star} + \max_{R_1, R_2 \in \mathcal{C}_k(\mathcal{R})} \left( \widehat{v}_{R_1}^{\widetilde{\pi}^{k,1}} - \widehat{v}_{R_2}^{\widetilde{\pi}^{k,1}} + \widehat{v}_{R_2}^{\widetilde{\pi}^{k,2}} - \widehat{v}_{R_1}^{\widetilde{\pi}^{k,2}} \right)$$

$$\leq \mathcal{E}_{\text{off}} + \max_{R_1, R_2 \in \mathcal{C}_k(\mathcal{R})} \left( \left( V_{R_1}^{\widetilde{\pi}^{k,1}} - V_{R_2}^{\widetilde{\pi}^{k,1}} + V_{R_2}^{\widetilde{\pi}^{k,2}} - V_{R_1}^{\widetilde{\pi}^{k,2}} \right) + \left( \widehat{v}_{R_1}^{\widetilde{\pi}^{k,1}} - V_{R_1}^{\widetilde{\pi}^{k,1}} \right) \right.$$

$$\left. + \left( \widehat{v}_{R_2}^{\widetilde{\pi}^{k,1}} - V_{R_2}^{\widetilde{\pi}^{k,1}} \right) + \left( \widehat{v}_{R_2}^{\widetilde{\pi}^{k,2}} - V_{R_2}^{\widetilde{\pi}^{k,2}} \right) + \left( \widehat{v}_{R_1}^{\widetilde{\pi}^{k,2}} - V_{R_1}^{\widetilde{\pi}^{k,2}} \right) \right)$$

$$\leq \mathcal{E}_{\text{off}} + \max_{R_1, R_2 \in \mathcal{C}_k(\mathcal{R})} \left( \left( V_{R_1}^{\widetilde{\pi}^{k,1}} - V_{R_2}^{\widetilde{\pi}^{k,1}} + V_{R_2}^{\widetilde{\pi}^{k,2}} - V_{R_1}^{\widetilde{\pi}^{k,2}} \right) + 4\mathcal{E}_{\text{off}} \right)$$

$$= 5\mathcal{E}_{\text{off}} + \max_{R_1, R_2 \in \mathcal{C}_k(\mathcal{R})} \left( V_{R_1}^{\widetilde{\pi}^{k,1}} - V_{R_2}^{\widetilde{\pi}^{k,1}} + V_{R_2}^{\widetilde{\pi}^{k,2}} - V_{R_1}^{\widetilde{\pi}^{k,2}} \right). \tag{29}$$

Consider the online preference-based regret as

$$\text{Reg}(K) := \frac{1}{2} \sum_{k=1}^{K} \left( V^{\pi^\star} - V^{\widetilde{\pi}^{k,1}} + V^{\pi^\star} - V^{\widetilde{\pi}^{k,2}} \right), \tag{30}$$

we have

$$\text{Reg}(K) \tag{31}$$

$$\leq \sum_{k=1}^{K} \max_{R_1, R_2 \in \mathcal{C}_k(\mathcal{R})} \left( V_{R_1}^{\widetilde{\pi}^{k,1}} - V_{R_2}^{\widetilde{\pi}^{k,1}} + V_{R_2}^{\widetilde{\pi}^{k,2}} - V_{R_1}^{\widetilde{\pi}^{k,2}} \right) + 5K\mathcal{E}_{\text{off}}$$

$$= \sum_{k=1}^{K} \max_{R_1, R_2 \in \mathcal{C}_k(\mathcal{R})} \left\{ \left( V_{R_1}(\tau^{k,1}) - V_{R_2}(\tau^{k,1}) + V_{R_2}(\tau^{k,2}) - V_{R_1}(\tau^{k,2}) \right) + \right.$$

$$+ (V_{R_1}^{\widetilde{\pi}^{k,1}} - V_{R_1}(\tau^{k,1})) - (V_{R_2}^{\widetilde{\pi}^{k,1}} - V_{R_2}(\tau^{k,1}))$$

$$\left. + (V_{R_1}^{\widetilde{\pi}^{k,2}} - V_{R_1}(\tau^{k,2})) - (V_{R_2}^{\widetilde{\pi}^{k,2}} - V_{R_2}(\tau^{k,2})) \right\} + 5K\mathcal{E}_{\text{off}}$$

$$\leq \sum_{k=1}^{K} \max_{R_1, R_2 \in \mathcal{C}_k(\mathcal{R})} \left( V_{R_1}(\tau^{k,1}) - V_{R_2}(\tau^{k,1}) + V_{R_2}(\tau^{k,2}) - V_{R_1}(\tau^{k,2}) \right)$$

$$+ 16\sqrt{\frac{K}{1-\gamma} \log\left(\frac{4}{\delta}\right)} + 5K\mathcal{E}_{\text{off}}$$

$$= \sum_{k=1}^{K} \max_{R_1, R_2 \in \mathcal{C}_k(\mathcal{R})} \left( (R_1(\tau^{k,1}) - R_1(\tau^{k,2})) - (R_2(\tau^{k,1}) - R_2(\tau^{k,2})) \right) \tag{32}$$

$$+ 16\sqrt{\frac{K}{1-\gamma} \log\left(\frac{4}{\delta}\right)} + 5K\mathcal{E}_{\text{off}}$$

$$\leq c_1 \sqrt{\kappa d_{\Delta\mathcal{R}} K \log\left(K|\Delta\mathcal{R}|/\delta\right)} + 16\sqrt{\frac{K}{1-\gamma} \log\left(\frac{4}{\delta}\right)} + 5K\mathcal{E}_{\text{off}}. \tag{33}$$

The first inequality follows from Equation 29. The second inequality follows from Azuma-Hoeffding's inequality and the fact that $V_R(\tau) - V_R^\pi$ is a martingale when $\tau \sim \pi$. Please refer to Cai et al. (2020) for a detailed derivation. The last inequality follows directly from Lemma 12.

Finally, set $\delta = 1/K$ and follow a standard argument for regret to PAC conversion (Jin et al., 2018), we can show that the expected suboptimality of average policy $\bar{\pi}$ generated by Algorithm 2 is upper bounded by

$$\text{SubOpt}(\bar{\pi}) \leq c_0 \cdot \sqrt{\frac{C^\dagger \log(N|\mathcal{V}||\Pi|)}{N(1-\gamma)^2}} + c_1 \cdot \sqrt{\frac{d_{\text{Elu}}(\Delta\mathcal{R}, 1/K) \log\left(K|\Delta\mathcal{R}|\right)}{K(1-\gamma)}}.$$

□

**Theorem 9** (Performance Guarantees with Pure Offline Queries). *Suppose (1) $Q^\star \in \mathcal{Q}, \pi^\star \in \Pi$, and (2) $\mathbb{T}^\pi q \in \mathcal{Q}, \forall \pi \in \Pi, q \in \mathcal{Q}$. Also we suppose the difference of return functions has a finite Eluder dimension $d_{Elu}(\Delta R, \alpha)$ and the underlying distribution of the offline dataset admits a finite coverage coefficient $C^\dagger$. Let $\beta_k = c_1 \sqrt{\log(K|\Delta\mathcal{R}|)/K}$ and $\epsilon = c_2 \sqrt{\log(N|\Pi||\mathcal{Q}|)/N}$, where $c_1, c_2$ are universal constants. Then the expected suboptimality of $\bar\pi$ from Algorithm 2 with pure offline queries is upper bounded by*

$$\text{SubOpt}(\bar\pi) \leq \mathcal{O}\left( \sqrt{\frac{C^\dagger \log(N|\mathcal{Q}||\Pi|)}{N(1-\gamma)^2}} + \sqrt{\frac{d_{Elu}(\Delta\mathcal{R}, 1/K)\log(K|\Delta\mathcal{R}|)}{K(1-\gamma)}} + \sqrt{\frac{C \log(N|\Delta R|)}{N(1-\gamma)}} \right),$$

(34)

*where $N$ is the size of the offline dataset, $K$ is the number of queries and $C = \max_s \frac{d^{\pi^*}(s)}{\mu(s)}$, where $\mu$ is the distribution that generates the dataset $\mathcal{D}$.*

*Proof.* The main difference between using pure offline queries and using online queries is that we have to use trajectories sampled from the dataset $\hat\tau^{k,1}, \hat\tau^{k,2}$ instead of online sampled trajectories $\tau^{k,1}, \tau^{k,2}$. This incurs an additional performance gap of $\mathcal{E}_{\text{gap}} = \sqrt{\frac{C \log(N|\Delta R|)}{N(1-\gamma)}}$, since we need to refine our query policies within the covered policy set

$$\Pi_{\text{covered}} = \left\{ \pi \mid \max_s \frac{d^\pi(s)}{\mu(s)} \leq C \right\}.$$

The proof for $\mathcal{E}_{\text{gap}}$ is the same as standard offline guarantees, and are omitted for simplicity. Then similar to the proof of Theorem 6, the regret can be bounded as

$$\begin{aligned}
\text{Reg}&(K) \\
&\leq \sum_{k=1}^K \max_{R_1, R_2 \in \hat{\mathcal{C}}_k(\mathcal{R})} \left( (R_1(\hat\tau^{k,1}) - R_1(\hat\tau^{k,2})) - (R_2(\hat\tau^{k,1}) - R_2(\hat\tau^{k,2})) \right) \\
&\quad + K\mathcal{E}_{\text{gap}} + 16\sqrt{\frac{K}{1-\gamma}\log\left(\frac{4}{\delta}\right)} + 5K\mathcal{E}_{\text{off}}.
\end{aligned}$$

(35)

Then following the proof of Theorem 6, we have

$$\text{SubOpt}(\bar\pi) \leq c_0 \cdot \sqrt{\frac{C^\dagger \log(N|\mathcal{V}||\Pi|)}{N(1-\gamma)^2}} + c_1 \cdot \sqrt{\frac{d_{Elu}(\Delta\mathcal{R}, 1/K)\log(K|\Delta\mathcal{R}|)}{K(1-\gamma)}} + c_2 \cdot \sqrt{\frac{C \log(N|\Delta R|)}{N(1-\gamma)}}.$$

$\square$

## C  Performance Guarantees with Pure Offline Queries

In pure offline settings, we have the following theorem.

**Theorem 10** (Performance Guarantees with Pure Offline Queries). *Suppose (1) $Q^\star \in \mathcal{Q}, \pi^\star \in \Pi$, and (2) $\mathbb{T}^\pi q \in \mathcal{Q}, \forall \pi \in \Pi, q \in \mathcal{Q}$. Also we suppose the difference of return functions has a finite Eluder dimension $d_{Elu}(\Delta R, \alpha)$ and the underlying distribution of the offline dataset admits a finite coverage coefficient $C^\dagger$. Let $\beta_k = c_1 \sqrt{\log(K|\Delta\mathcal{R}|)/K}$ and $\epsilon = c_2 \sqrt{\log(N|\Pi||\mathcal{Q}|)/N}$, where $c_1, c_2$ are universal constants. Then the expected suboptimality of $\bar{\pi}$ from Algorithm 2 with pure offline queries is upper bounded by*

$$\text{SubOpt}(\bar{\pi}) \leq \mathcal{O}\left( \sqrt{\frac{C^\dagger \log(N|\mathcal{Q}||\Pi|)}{N(1-\gamma)^2}} + \sqrt{\frac{d_{Elu}(\Delta\mathcal{R}, 1/K) \log(K|\Delta\mathcal{R}|)}{K(1-\gamma)}} + \sqrt{\frac{C \log(N|\Delta R|)}{N(1-\gamma)}} \right),$$
(36)

*where $N$ is the size of the offline dataset, $K$ is the number of queries and $C = \max_s \frac{d^{\pi^*}(s)}{\mu(s)}$, where $\mu$ is the distribution that generates the dataset $\mathcal{D}$.*

*Proof.* The main difference between using pure offline queries and using online queries is that we have to use trajectories sampled from the dataset $\hat{\tau}^{k,1}, \hat{\tau}^{k,2}$ instead of online sampled trajectories $\tau^{k,1}, \tau^{k,2}$. This incurs an additional performance gap of $\mathcal{E}_{\text{gap}} = \sqrt{\frac{C \log(N|\Delta R|)}{N(1-\gamma)}}$, since we need to refine our query policies within the covered policy set

$$\Pi_{\text{covered}} = \left\{ \pi \mid \max_s \frac{d^\pi(s)}{\mu(s)} \leq C \right\}.$$

The proof for $\mathcal{E}_{\text{gap}}$ is the same as standard offline guarantees, and are omitted for simplicity. Then similar to the proof of Theorem 6, the regret can be bounded as

$$\text{Reg}(K) \leq \sum_{k=1}^{K} \max_{R_1, R_2 \in \widehat{\mathcal{C}}_k(\mathcal{R})} \left( (R_1(\hat{\tau}^{k,1}) - R_1(\hat{\tau}^{k,2})) - (R_2(\hat{\tau}^{k,1}) - R_2(\hat{\tau}^{k,2})) \right)$$

$$+ K\mathcal{E}_{\text{gap}} + 16\sqrt{\frac{K}{1-\gamma} \log\left(\frac{4}{\delta}\right)} + 5K\mathcal{E}_{\text{off}}.$$
(37)

Then following the proof of Theorem 6, we have

$$\text{SubOpt}(\bar{\pi}) \leq c_0 \cdot \sqrt{\frac{C^\dagger \log(N|\mathcal{V}||\Pi|)}{N(1-\gamma)^2}} + c_1 \cdot \sqrt{\frac{d_{Elu}(\Delta\mathcal{R}, 1/K) \log(K|\Delta\mathcal{R}|)}{K(1-\gamma)}} + c_2 \cdot \sqrt{\frac{C \log(N|\Delta R|)}{N(1-\gamma)}}.$$

$$\square$$

# D  AUXILIARY LEMMAS

**Lemma 11.** *With probability at least $1 - \delta$, the following event $\mathcal{E}_1$ holds*

$$R^\star \in \mathcal{C}_k(\mathcal{R}), \quad \forall k \in [K],$$

*where*

$$\mathcal{C}_k(\mathcal{R}) = \left\{ R \in \mathcal{R} : ((\widehat{R}(\tau_1) - \widehat{R}(\tau_2)) - (R(\tau_1) - R(\tau_2)))^2 \leq c\kappa \log(K|\Delta\mathcal{R}|/\delta) \right\},$$

*$c$ is an absolute constant and $\kappa := \frac{1}{\sigma'(2R_{max})}$ is the degree of non-linearity of the link function $\sigma$.*

*Proof.* Using Lemma 16, we have that

$$\sum_{i=1}^k \left\| \mathbb{P}(\tau_k^1 \succ \tau_k^2 | \widehat{R}) - \mathbb{P}(\tau_k^1 \succ \tau_k^2 | R^\star) \right\|_{\text{TV}}^2 \leq 2 \log(|\Delta\mathcal{R}|/\delta).$$

Note that $\mathbb{P}(\tau_k^1 \succ \tau_k^2 | R) = \sigma(R(\tau_1) - R(\tau_2))$, and $R(\tau)$ is bounded by $R_{\max}$, we have

$$\sum_{i=1}^k ((\widehat{R}(\tau_1) - \widehat{R}(\tau_2)) - (R^\star(\tau_1) - R^\star(\tau_2)))^2 \leq c\kappa \log(|\Delta\mathcal{R}|/\delta)$$

Then, by the union bound, we have the conclusion immediately. $\qquad\square$

**Lemma 12.** *Under event $\mathcal{E}_1$ in Lemma 11, it holds that*

$$\sum_{k=1}^K \left| (R_1(\tau^{k,1}) - R_1(\tau^{k,2})) - (R_2(\tau^{k,1}) - R_2(\tau^{k,2})) \right| \leq O\left( \sqrt{d_{Elu}(\Delta\mathcal{R}, \delta) K \log(K|\Delta\mathcal{R}|/\delta)} \right). \tag{38}$$

*Proof.* Under event $\mathcal{E}_1$, we have $\max_{1 \leq k \leq K} \text{diam}(\mathcal{B}_{(\tau_1, \tau_2)_{1:k}}(\mathcal{C}_k(\mathcal{R}))) \leq 2\sqrt{\kappa \log(K|\Delta\mathcal{R}|/\delta)}$ by Lemma 16, where

$$\mathcal{B}_{(\tau_1, \tau_2)_{1:k}}(\mathcal{F}) := \sup_{f_1, f_2 \in \mathcal{F}} \left( \sum_{t=1}^k ((f_1(\tau_1^t) - f_1(\tau_2^t)) - (f_2(\tau_1^t) - f_2(\tau_2^t)))^2 \right)^{1/2}.$$

Therefore, following Lemma 13, we have

$$\begin{aligned}
&\sum_{k=1}^K \left| (R_1(\tau^{k,1}) - R_1(\tau^{k,2})) - (R_2(\tau^{k,1}) - R_2(\tau^{k,2})) \right| \\
&\leq \sum_{k=1}^K \mathcal{B}_{(\tau_1^k, \tau_2^k)}(\mathcal{R}_k) \\
&\leq O\left( \sqrt{d_{\text{Elu}}(\Delta\mathcal{R}, \delta) K \log(K|\Delta\mathcal{R}|/\delta)} \right).
\end{aligned} \tag{39}$$

$\square$

**Lemma 13** (Lemma 5 of Russo & Van Roy (2014)). *. Let $\mathcal{V} \in \mathcal{B}_\infty(\mathcal{X}, C)$ be a set of functions bounded by $C > 0$, $(\mathcal{V}_t)_{t \geq 1}$ and $(x_t)_{t \geq 1}$ be sequences such that $\mathcal{V}_t \subseteq \mathcal{V}$ and $x_t \in \mathcal{X}$ hold for $t \geq 1$. Let $\mathcal{V}|_{x_{1:t}} = \{(f(x_1), \ldots, f(x_t)) : f \in \mathcal{V}\}(\subseteq \mathbb{R}^t)$ and for $S \subseteq \mathbb{R}$, let $\text{diam}(S) = \sup_{u,v \in S} \|u - v\|_2$ be the diameter of $S$. Then, for any $T \geq 1$ and $\alpha > 0$ it holds that*

$$\sum_{t=1}^T \text{diam}(\mathcal{V}_t|_{x_t}) \leq \alpha + C(d \wedge T) + 2\delta_T \sqrt{dT}, \tag{40}$$

*where $\delta_T = \max_{1 \leq t \leq T} \text{diam}(\mathcal{V}_t|_{x_{1:t}})$ and $d = \dim_\epsilon(\mathcal{V}, \alpha)$.*

The following lemmas summarizes the results regarding General Function Estimator.

**Lemma 14** (Theorem A.1 in Xie et al. (2021)). *For any $\pi \in \Pi$, let $q_\pi$ be defined as follows,*

$$q_\pi := \arg\min_{q \in \mathcal{Q}} \sup_{\text{admissible } \nu} \|q - T^\pi q\|^2_{2,\nu}. \tag{41}$$

*Then the following event $\mathcal{E}_2$ holds with probability as least $1 - \delta$:*

$$\mathcal{E}(q_\pi, \pi; \mathcal{D}) \leq \frac{139 \log \frac{|\mathcal{Q}||\Pi|}{\delta}}{n(1-\gamma)}, \tag{42}$$

*where $\mathcal{E}(q, \pi; \mathcal{D}) := \mathcal{L}(q, q, \pi; \mathcal{D}) - \min_{q' \in \mathcal{V}} \mathcal{L}(q', q, \pi; \mathcal{D})$.*

The following lemma shows that $\mathcal{E}(q, \pi; \mathcal{D})$ could effectively estimate $\|q - T^\pi q\|^2_{2,\mu}$.

**Lemma 15** (Theorem A.2 in Xie et al. (2021)). *For any $\pi \in \Pi, q \in \mathcal{Q}, h \in [H]$, and any $\epsilon > 0$, if $\mathcal{E}(q, \pi; \mathcal{D}) \leq \epsilon$, Then the following event $\mathcal{E}_3$ holds with probability as least $1 - \delta$:*

$$\|q - T^\pi q'\|_{2,\mu} \leq \sqrt{\frac{231 \log \frac{|\mathcal{Q}||\Pi|}{\delta}}{n(1-\gamma)}} + \sqrt{\epsilon} := \epsilon_b. \tag{43}$$

**Lemma 16** (Theorem 21 in Agarwal et al. (2020)). *Fix $\delta \in (0, 1)$, assume $|\mathcal{F}| < \infty$ and $f^\star \in \mathcal{F}$. Then with probability at least $1 - \delta$*

$$\sum_{i=1}^{n} \mathbb{E}_{x \sim \mathcal{D}_i} \left\| \widehat{f}(x, \cdot) - f^\star(x, \cdot) \right\|^2_{TV} \leq 2 \log(|\mathcal{F}|/\delta).$$

# E    ADDITIONAL EXPERIMENTAL RESULTS

**Experiments on Meta-World**    Table 5 shows the complete experimental results in Meta-World.

| Task | OPRL | PT | PT+PDS | IDRL | OPRIDE |
|---|---|---|---|---|---|
| assembly-v2 | 10.1±0.5 | 10.2±0.7 | 12.8±0.6 | 10.3±1.9 | **14.2±1.3** |
| basketball-v2 | 11.7±10.2 | 80.7±0.1 | 78.7±2.0 | **82.7±2.5** | 61.4±2.3 |
| bin-picking-v2 | 82.0±5.6 | 31.9±16.2 | 53.4±19.0 | 84.7±2.9 | **93.3±3.2** |
| button-press-wall-v2 | 51.7±1.6 | 58.8±0.9 | 59.4±0.9 | 69.0±1.0 | **77.7±0.1** |
| box-close-v2 | 15.0±0.7 | **17.7±0.1** | 17.2±0.3 | 16.9±0.6 | 16.8±0.4 |
| coffee-push-v2 | 1.7±1.7 | 1.3±0.5 | 1.3±0.5 | 42.0±3.8 | **59.4±24.8** |
| disassemble-v2 | 8.4±0.8 | 6.0±0.4 | 7.6±0.2 | 7.4±1.9 | **26.6±4.9** |
| door-close-v2 | 61.2±1.3 | 65.1±10.1 | 62.4±8.7 | 78.1±3.2 | **94.8±1.1** |
| door-unlock-v2 | **79.2±2.3** | 73.7±5.4 | 73.6±4.8 | 71.2±2.9 | 71.0±2.3 |
| drawer-open-v2 | 53.0±3.3 | 59.7±1.3 | 58.3±0.1 | 62.5±2.0 | **68.7±3.0** |
| faucet-close-v2 | 60.8±1.0 | 57.8±0.9 | 46.2±0.2 | 61.5±3.2 | **73.1±0.8** |
| hammer-v2 | 16.4±1.0 | 30.2±1.7 | 32.6±0.8 | 33.6±2.8 | **50.3±3.2** |
| hand-insert-v2 | 5.2±3.2 | 18.7±0.1 | 20.3±0.6 | 41.9±2.7 | **61.8±4.9** |
| handle-press-v2 | **28.7±4.0** | 27.9±0.2 | 28.2±0.2 | 28.0±0.4 | **28.7±0.1** |
| lever-pull-v2 | **63.2±10.4** | 49.2±3.7 | 51.7±0.1 | 33.1±1.2 | 51.8±1.6 |
| peg-insert-side-v2 | 3.5±1.8 | 16.8±0.1 | 12.4±1.4 | 67.4±0.1 | **79.0±0.2** |
| plate-slide-v2 | 77.4±1.6 | 4.9±0.0 | 37.3±2.3 | **79.6±3.5** | 79.9±4.6 |
| push-v2 | 10.6±1.5 | 16.7±5.0 | 1.8±0.4 | 30.7±5.3 | **59.1±5.4** |
| push-back-v2 | 0.8±0.0 | 1.1±0.4 | 1.1±0.1 | 14.0±1.1 | **17.7±2.0** |
| push-wall-v2 | 7.4±4.2 | 74.8±14.4 | 3.4±0.9 | 89.2±3.2 | **102.2±1.2** |
| reach-v2 | 63.5±2.9 | 82.0±0.8 | 84.3±0.9 | 75.8±1.8 | **88.0±0.5** |
| soccer-v2 | 34.3±4.0 | **51.3±4.1** | 41.5±11.9 | 44.3±2.1 | 45.4±3.9 |
| sweep-into-v2 | 37.1±13.9 | 9.8±0.2 | 9.2±0.1 | 63.1±3.5 | **71.6±0.1** |
| sweep-v2 | 6.8±1.8 | 8.0±0.4 | 8.0±0.1 | 73.0±2.8 | **78.5±1.0** |

Table 5: Performance of offline RL algorithm on the reward-labeled dataset with different preference reward learning methods on the Meta-World tasks.

**Experiments on Mujoco and Kitchen**    We conduct a wider range of experiments on MuJoCo and Kitchen tasks. The experimental results in Table 6 show that OPRIDE achieves superior performance compared with other baselines. The experimental results also demonstrate that the In-Dataset Exploration and Variance-based Discount Scheduling mechanisms we proposed can be effectively applied to different tasks.

| Domain | Tasks | OPRL | PT | PT+PDS | OPRIDE |
|---|---|---|---|---|---|
| Mujoco | hopper-medium | 23.0±0.1 | 36.9±2.1 | 35.8±1.8 | **38.5±2.2** |
| | hopper-medium-expert | 57.7±23.7 | 68.0±2.6 | 69.1±1.7 | **92.3±15.8** |
| | walker2d-medium | 70.6±1.1 | **71.7±2.6** | 70.9±1.8 | **72.7±1.8** |
| | walker2d-medium-expert | 108.3±3.8 | **109.4±0.3** | 108.4±0.5 | **110.3±0.2** |
| | halfcheetah-medium | 41.9±0.1 | **42.1±0.1** | 41.5±0.1 | **42.4±0.1** |
| | halfcheetah-medium-expert | 81.8±0.6 | 81.9±0.1 | 82.4±0.2 | **86.5±1.5** |
| | kitchen-partial | 34.6±0.2 | 48.2±4.1 | **51.1±2.3** | 38.7±3.7 |
| | kitchen-mixed | 46.9±0.1 | 42.5±1.0 | 44.9±1.9 | **49.8±0.1** |
| | kitchen-partial | **62.6±1.7** | 47.5±2.5 | 49.8±4.5 | **63.7±1.1** |

Table 6: Performance of offline RL algorithm on the reward-labeled dataset with different preference reward learning methods on the Mujoco tasks.

**Ablation about In-Dataset Exploration module**    The choice to emphasize value functions over reward functions is crucial due to their ability to guide policy optimization effectively. Intuitively, while maximizing the information gain concerning the reward function (e.g., difference over the

| $\gamma_{small}$ | 0.5 | 0.6 | 0.7 | 0.8 | 0.9 | 0.95 |
|---|---|---|---|---|---|---|
| bin-picking | 72.1±23.9 | 87.8±2.7 | 93.3±3.2 | 84.6±9.4 | 74.8±33.5 | 70.9±9.4 |
| button-press-wall | 77.6±0.3 | 77.4±0.3 | 77.7±0.1 | 71.0±0.7 | 69.2±0.9 | 68.1±9.7 |
| door-close | 88.4±0.8 | 89.9±0.7 | 94.8±1.1 | 90.0±2.1 | 91.1±1.5 | 87.6±0.7 |
| faucet-close | 58.1±5.2 | 58.2±12.5 | 73.1±0.8 | 61.4±2.7 | 55.1±3.7 | 57.4±12.1 |

Table 7: Performance of offline RL algorithm on the reward-labeled dataset with various discount factor values $\gamma_{small}$ on the high variance data points.

reward function) can help learn a well-calibrated reward function, it can still be sample inefficient in determining the optimal policy since we are not interested in the accuracy of the reward function in low-return regions. For instance, suppose we have actions $a_1$ and $a_2$ that lead to a terminal state $s_0$, and their immediate rewards are highly uncertain, ranging from $[-1, 1]$. And we have actions $a_3$ and $a_4$ that lead to high return states $s_1$ but yield a known fixed immediate reward of zero. By maximizing the reward differences, we will compare $a_1$ and $a_2$, but such comparison contains no information in determining the optimal policy, which will not choose $a_1$ and $a_2$ at all. Theoretically, maximizing the information gain with respect to the reward function is insufficient to derive a performance guarantee for PbRL.

We conduct additional ablation studies for these two mechanisms. The experimental results in Table 8 show that maximizing information gain about the optimal policy can achieve better performance than the reward function.

| Domain | Tasks | OPRIDE (Reward Difference) | OPRIDE (Value Function Difference) |
|---|---|---|---|
| | bin-picking | 78.5±17.8 | **93.3±3.2** |
| | button-press-wall | 67.4±5.4 | **77.7±0.1** |
| | door-close | 88.3±0.7 | **94.8±1.1** |
| Metaworld | faucet-close | 48.7±0.6 | **73.1±0.8** |
| | peg-insert-side | 9.7±8.5 | **79.0±0.2** |
| | reach | **86.6±0.1** | 88.0±0.5 |
| | sweep | 18.2±2.9 | **78.5±1.0** |

Table 8: Ablation study on the metaworld tasks.

**Ablation about Variance-based Discount Scheduling module**   The choice of using a pessimistic discount factor in offline RL draws on theoretical guarantees discussed in prior works (Jiang et al., 2015; Hu et al., 2022). While prior methods may utilize a smaller fixed discount factor (Jiang et al., 2015) or tuned values in imitation learning (Liu et al., 2023), our approach innovatively employs variance-based discount scheduling to mitigate reward overestimation issues specific to offline Preference-based RL.

A smaller discount factor serves a dual purpose: it regulates optimality against sample efficiency trade-offs (Hu et al., 2022) and aligns with model-based pessimism principles, ensuring robust policy learning. Conversely, multiplicative adjustments to rewards lack theoretical grounding and often yield suboptimal performance, as evidenced in Table 9.

| Domain | Tasks | OPRIDE (Penalise Reward) | OPRIDE (Penalise Discount Factor) |
|---|---|---|---|
| | bin-picking | 53.4±19.0 | **93.3±3.2** |
| | button-press-wall | 59.4±0.9 | **77.7±0.1** |
| | door-close | 62.4±8.7 | **94.8±1.1** |
| Metaworld | faucet-close | 46.2±0.2 | **73.1±0.8** |
| | peg-insert-side | 12.4±1.4 | **79.0±0.2** |
| | reach | 84.3±0.9 | **88.0±0.5** |
| | sweep | 8.0±0.1 | **78.5±1.0** |

Table 9: Ablation studies about penalizing rewards and the discount factor.

**Ablation for hyperparameter** $m$    We conduct an ablation study on the hyperparameter $m$, with the results presented in Table 10. Our findings indicate a clear trade-off. An excessively large $m$ reduces the discount factor $\gamma$ for too many data points, leading to an overly pessimistic value estimation. Conversely, a value of $m$ that is too small provides an insufficient penalty for estimation uncertainty. Based on this, we recommend using a larger $m$ for more complex tasks (which tend to have higher estimation noise) and a smaller $m$ for simpler tasks.

| $m\%$ | 10% | 20% | 30% | 40% | 50% |
|---|---|---|---|---|---|
| bin-picking | 71.4±3.3 | 85.9±3.9 | 93.3±3.2 | 88.7±2.6 | 73.2±2.7 |
| button-press-wall | 67.2±2.1 | 70.2±0.6 | 77.7±0.1 | 77.5±0.3 | 77.5±0.6 |
| door-close | 87.9±1.2 | 89.6±1.0 | 94.8±1.1 | 89.6±0.8 | 87.3±0.9 |
| faucet-close | 56.8±2.6 | 62.3±1.6 | 73.1±0.8 | 58.7±1.9 | 57.6±2.1 |

Table 10: Ablation study for the hyperparameter $m$ from 10% to 50%.

**Softer confidence discount mechanism**    We investigate a more adaptive, "soft" confidence discount mechanism and compared it with our current threshold-based approach. Specifically, we implement a continuous annealing strategy where the smaller discount factor, $\gamma_{\text{small}}$, is adjusted based on the variance of the ensemble's value function estimates, $\text{Var}[Q_{\phi_i}(s,a)]_{i=1}^{M}$. In this setup, $\gamma_{\text{small}}$ decreases as the variance increases, governed by the formula:

$$\gamma_{\text{small}} = \frac{\gamma}{\max(1, \alpha \cdot \text{Var}[Q_{\phi_i}(s,a)]_{i=1}^{M})}.$$

The experimental results, presented in Table 11, show that the performance of this continuous annealing approach is comparable to our hard-penalty method. Given its comparable performance and simpler implementation, we opted for the threshold-based approach in our final model.

| Tasks | Continuous annealing | Threshold-based |
|---|---|---|
| bin-picking | 94.1±3.7 | 93.3±3.2 |
| button-press-wall | 77.4±0.3 | 77.7±0.1 |
| door-close | 94.3±1.7 | 94.8±1.1 |
| faucet-close | 71.6±0.9 | 73.1±0.8 |
| peg-insert-side | 80.5±0.3 | 79.0±0.2 |
| reach | 87.5±0.6 | 88.0±0.5 |
| sweep | 79.8±1.2 | 78.5±1.0 |

Table 11: Ablation study for the discount factor mechanism on the Meta-World tasks.

**Computational cost**    We present the computational cost of OPRIDE on the GeForce RTX 3090 GPU device, as shown in Table 12. The reported time is the sum of the reward function training time and the policy training time. The experimental results indicate that as the number of ensembles $M$ increases, the computational cost does not increase significantly. This is because we have adopted a multi-head mechanism instead of separate training, thereby saving training time.

| $M$ | 2 | 5 | 10 |
|---|---|---|---|
| bin-picking | 1.1h | 1.2h | 1.3h |
| button-press-wall | 1.1h | 1.1h | 1.2h |
| door-close | 1.1h | 1.2h | 1.3h |
| faucet-close | 0.9h | 1.0h | 1.2h |
| peg-insert-side | 1.0h | 1.1h | 1.3h |

Table 12: Computational Cost for OPRIDE with ensemble $M$, where $h$ is the hour. The report time is the sum of the reward function training time and the policy training time.

**Ablation study of query number**  We conduct experiments with various query numbers. The experimental results in Table 13 show that performance gains accelerate most rapidly between 0.1% and 0.2% of queries. Beyond 0.3%, improvements gradually taper off, ultimately stabilizing at around 1%. At this point, adding further queries yields only marginal performance benefits.

| Query Number | 1 (BC) | 10 (0.1%) | 13 | 20 (0.2%) | 30 (0.3%) | 50 (0.5%) | 100 (1.0%) |
|---|---|---|---|---|---|---|---|
| Hammer-v2 | 13.2±3.3 | 50.3±3.2 | 55.4±2.6 | 75.9±2.2 | 79.1±2.6 | 79.5±2.7 | 79.9±2.3 |
| Coffee-push-v2 | 2.5±1.1 | 56.7±2.8 | 57.6±2.5 | 55.3±3.1 | 58.9±3.7 | 59.4±3.5 | 59.5±3.4 |
| Disassemble-v2 | 9.3±1.7 | 26.6±2.8 | 27.8 ± 3.1 | 30.6±5.2 | 35.7±4.8 | 37.2±4.6 | 38.3±5.1 |
| Push-v2 | 1.2±0.6 | 50.9±1.8 | 63.7±2.1 | 76.8±2.4 | 82.7±2.6 | 83.6±2.3 | 84.0 ±2.2 |

Table 13: Performance of OPRIDE with various query numbers.

**Comparison with one-stage and sequential ranked list methods**  We compare our method with one-stage framework (IPL (Hejna & Sadigh, 2023), CPL (Hejna et al.)) and sequential ranked list method (LiRE) (Choi et al., 2024). The experimental results in Table 14 show that compared to PT, one-stage framework and LiRE achieve better performance by either bypassing reward modeling or enhancing query sampling mechanisms. Meanwhile, OPRIDE outperforms all of them, demonstrating that our iterative two-stage framework can explore more valuable queries.

| Tasks | PT | IPL | CPL | LiRE | OPRIDE |
|---|---|---|---|---|---|
| lever-pull | 49.2±3.7 | 50.2±2.1 | 50.1±2.5 | 51.2±1.8 | 51.8±1.6 |
| peg-insert-side | 16.8±0.1 | 53.8±0.4 | 54.7±0.3 | 63.1±0.2 | 79.0±0.2 |
| plate-slide | 4.9±0.0 | 60.9±5.6 | 61.7±4.8 | 68.3±4.3 | 79.9±4.6 |
| push | 16.7±5.0 | 38.5±4.2 | 39.2±3.9 | 45.5±3.7 | 59.1±5.4 |
| push-back | 1.1±0.4 | 9.8±1.6 | 9.3±1.8 | 13.2±1.5 | 17.7±2.0 |
| push-wall | 74.8±14.4 | 85.8±4.7 | 87.3±3.8 | 90.2±3.6 | 102.2±1.2 |
| reach | 82.0±0.8 | 84.2±0.2 | 84.7±0.6 | 85.4±0.3 | 88.0±0.5 |
| soccer | 51.3±4.1 | 52.3±4.7 | 53.6±3.9 | 54.2±3.6 | 45.4±3.9 |
| sweep-into | 9.8±0.2 | 54.2±0.3 | 56.8±0.3 | 62.9±0.2 | 71.6±0.1 |
| sweep | 8.0±0.4 | 69.3±1.1 | 68.2±1.3 | 71.3±1.6 | 78.5±1.0 |

Table 14: Comparison with one-stage framework and sequential ranked list method.

**Pixel-based and actual human-in-the-loop environments**  we conducted experiments on pixel-based Atari environments to rigorously test OPRIDE's applicability beyond vector-state domains. In addition, these experiments incorporated actual human preference feedback rather than synthetic scripted teachers. We recruited 15 human evaluators with prior gaming experience to provide trajectory comparisons through an intuitive interface. Each evaluator performed 50 pairwise comparisons per task, with queries selected by OPRIDE's exploration mechanism.

As shown in Table 15, OPRIDE achieves state-of-the-art results across all five Atari games. Notably, these improvements persist despite the added complexity of image-based state representations and inherent noise in human labeling. This validates OPRIDE's robustness to visual inputs and its effectiveness with genuine human feedback. We attribute this success to two key design features: (1) The exploration strategy's focus on segment-level value differences remains effective when states are represented as latent features from CNN encoders, and (2) The variance-based discount scheduling mitigates overoptimization risks amplified by noisy human labels.

| Tasks | OPRL | PT | PT+PDS | IDRL | OPRIDE |
|-------|------|-----|--------|------|--------|
| Pong | 9.6±1.4 | 9.4±1.0 | 8.5±1.8 | 15.3±1.1 | 17.8±1.3 |
| Breakout | 125.9±14.2 | 79.9±13.9 | 86.3±13.8 | 153.7±14.5 | 256.7±14.3 |
| Q*bert | 7924.6±376.1 | 7482.9±353.4 | 6844.2±394.6 | 8294.1±359.7 | 13535.2±327.2 |
| Seaquest | 2784.1±72.7 | 2538.3±78.9 | 2459.6±74.5 | 2941.2±69.2 | 3478.4±71.3 |
| Asterix | 164.9±21.5 | 155.8±24.6 | 146.3±27.4 | 357.4±30.1 | 426.9±28.7 |

Table 15: Experiments on pixel-based Atari tasks with human-in-the-loop.

## F    EXPERIMENT DETAILS

**Experimental Setup**    For the Meta-World tasks, each dataset consists of 1000 trajectories. 50 trajectories of which are collected by the corresponding scripted policy added with a Gaussian noise $\mathcal{N}(0, 0.8)$ to increase diversity, and the rest 950 trajectories are collected with a policy that is a $\epsilon$-greedy variant to the former noisy policy and select random actions with probability $\epsilon = 0.8$. For the Antmaze tasks, we use the standard dataset in the D4RL benchmark but remove the reward labels.

**OPRL**    We use the official implementation [*], which uses 7 ensembles. Each ensemble is initially trained with 1 randomly selected query and then performs 3 rounds of active querying and training, and in each round, 1 query is acquired, making a total of 10 queries.

**PT**    We use the official implementation [†]. We follow its original hyper-parameter settings, and change the number of queries to 10.

**IDRL**    We use the official implementation [‡]. We follow its original hyper-parameter settings.

**Survival Instinct**    We use the official implementation [§]. We follow its original hyper-parameter settings.

**OPRIDE**    Our code is built on PT. We use the same transformer architecture and hyper-parameter with PT. The ensemble number $N$ is 2. The size of $\mathcal{D}$ is 10000. The offline pre-training step for $V_i(\cdot, \cdot)$ in the Equation 7 is $10000 \times c$, where $c$ is the $c$-th selected query. Please refer to Table 16 for detailed parameters.

| Hyperparameter | Value |
| --- | --- |
| Optimizer | Adam |
| Critic learning rate | 3e-4 |
| Actor learning rate | 3e-4 |
| Mini-batch size | 256 |
| Discount factor | 0.99 |
| Target update rate | 5e-3 |
| IQL parameter $\tau$ | 0.7 |
| IQL parameter $\alpha$ | 3.0 |
| Query Number | 10 |
| **OPRL** | **Value** |
| Ensemble Number | 7 |
| **OPRIDE** | **Value** |
| Ensemble Number $N$ | 2 |
| Size of $\mathcal{D}$ | 10000 |
| Offline Pre-training step | $10000 \times c$ |
| Top $m\%$ | Top 30% |
| $\gamma_{small}$ | 0.7 |
| $S$ | 1000 |

Table 16: Hyper-parameters sheet of Algorithms.

[*] https://github.com/danielshin1/oprl
[†] https://github.com/csmile-1006/PreferenceTransformer/tree/main
[‡] https://github.com/david-lindner/idrl
[§] https://survival-instinct.github.io

