# OpenReview forum: "OPRIDE: Efficient Offline Preference-based Reinforcement Learning via In-Dataset Exploration"
_ICLR.cc/2026/Conference — ICLR 2026 Poster_

### Official Review · Reviewer_1iWt · 2025-10-23

**Soundness:** 3
**Presentation:** 3
**Contribution:** 2
**Rating:** 4
**Confidence:** 4

**Summary:**

The paper introduces OPRIDE, an offline PbRL framework designed to improve query efficiency and mitigate overoptimization. It comprises two key components: In-dataset exploration (IDE), which selects queries maximizing differences among value functions trained on bootstrapped reward models, and variance-based discount scheduling (VDS), which dynamically reduces the discount factor for high variance state-action pairs to mitigate overestimation and stabilize learning. The authors provide theoretical guarantees on query efficiency, and experiments on Meta-World and AntMaze benchmarks demonstrate that OPRIED achieves superior performance with only 10 queries compared to prior offline PbRL baselines.

**Strengths:**

1. The IDS mechanism is intuitive and well-motivated. Unlike disagreement-based methods that focus on reward uncertainty, IDS selects queries that maximize information gain about optimal policy by measuring value function difference. Table 3 empirically validates that IDS significantly outperforms random and disagreement-based selection.
2. The method achieves compelling performance with only 10 queries across diverse tasks (Tables 1 & 2, Figure 2), significantly outperforming baselines.
3. The paper provides thorough ablations examining IDS vs. baselines (Table 3), VDS effectiveness (Table 9), and sensitivity to hyperparameters (Tables 7, 10). These studies clearly demonstrate the contribution of each component.

**Weaknesses:**

1. This paper only compares two-stage methods such as OPRL and PT. However, recent advances in offline PbRL have introduced a distinct line of one-stage framework that learns policies without reward learning, including IPL [1] and CPL [2]. These approaches are generally more robust and often outperform two-stage reward learning frameworks, as they avoid the approximation errors introduced by learning a separate reward estimator. it does not suffer from approximation error in reward estimator learning. The authors should include comparisons with these one-stage methods to more convincingly demonstrate the superiority of the proposed approach.

2. Please also refer other two-stage offline PbRL studies, such as CLARIFY [3] and LiRE [4], and discuss how they related to the proposed method in the Related Works section.

3. Experiments with non-ideal and noisy annotator (real human) are missing. To validate the practicality of the proposed method in real-world scenarios, the authors should include experiments involving actual human feedback [3, 4].

4. The experiments are conducted only in vector-based environments. While most offline PbRL studies do not include experiments with proprioceptive (pixel-based state) inputs, evaluating such setting is essential for real-sensory and visual domains. Please refer to recent online PbRL studies that demonstrated effectiveness in pixel-based environments for this setting [5,6].

References

[1] Hejna, J., & Sadigh, D. (2023). Inverse preference learning: Preference-based rl without a reward function. Advances in Neural Information Processing Systems, 36, 18806-18827.

 [2] Hejna, J., Rafailov, R., Sikchi, H., Finn, C., Niekum, S., Knox, W. B., & Sadigh, D. Contrastive Preference Learning: Learning from Human Feedback without Reinforcement Learning. In The Twelfth International Conference on Learning Representations.

[3] Mu, N., Hu, H., Hu, X., Yang, Y., XU, B., & Jia, Q. S. CLARIFY: Contrastive Preference Reinforcement Learning for Untangling Ambiguous Queries. In Forty-second International Conference on Machine Learning.

[4] Choi, H., Jung, S., Ahn, H., & Moon, T. (2024, July). Listwise Reward Estimation for Offline Preference-based Reinforcement Learning. In International Conference on Machine Learning (pp. 8651-8671). PMLR.

[5] Park, J., Seo, Y., Shin, J., Lee, H., Abbeel, P., & Lee, K. SURF: Semi-supervised Reward Learning with Data Augmentation for Feedback-efficient Preference-based Reinforcement Learning. In International Conference on Learning Representations.

[6] Metcalf, K., Sarabia, M., Mackraz, N., & Theobald, B. J. (2023, December). Sample-Efficient Preference-based Reinforcement Learning with Dynamics Aware Rewards. In Conference on Robot Learning (pp. 1484-1532). PMLR.

[7] Van Hasselt, H., Guez, A., & Silver, D. (2016, March). Deep reinforcement learning with double q-learning. In Proceedings of the AAAI conference on artificial intelligence (Vol. 30, No. 1).

[8] Nikishin, E., Schwarzer, M., D’Oro, P., Bacon, P. L., & Courville, A. (2022, June). The primacy bias in deep reinforcement learning. In International conference on machine learning (pp. 16828-16847). PMLR.

[9] Coste, T., Anwar, U., Kirk, R., & Krueger, D. Reward Model Ensembles Help Mitigate Overoptimization. In The Twelfth International Conference on Learning Representations.

[10] Nauman, M., Bortkiewicz, M., Miłoś, P., Trzcinski, T., Ostaszewski, M., & Cygan, M. (2024, July). Overestimation, Overfitting, and Plasticity in Actor-Critic: the Bitter Lesson of Reinforcement Learning. In International Conference on Machine Learning (pp. 37342-37364). PMLR.

**Questions:**

1. How high variance in value estimation directly relates to overestimation? Overestimation can instead be measured explicitly as the gap between predicted Q-values and actual returns, or indirectly detecting spikes in streaming critic outputs [7,8].

2. If overoptimization and overestimation need to be mitigated, alternative strategies such as using minimum prediction among ensemble reward models, spectral normalization and periodic resets can serve as direct remedies [9, 10]. How does the proposed VDS compare to these existing techniques?

3. How would the proposed framework behave under noisy or inconsistent human feedback, which is common in real-world preference learning settings?

---

> ### Author Response · Authors · 2025-11-21
> **Response to Reviewer 1iWt (I)**
>
> Dear Reviewer,
>
> Thank you for your constructive and valuable comments. We have conducted additional experiments and analysis to address your concerns, and we believe our responses have strengthened the manuscript.
>
> **W1: Comparison with one-stage framework.**
>
> **A for W1:**
> Following your suggestion, we have benchmarked OPRIDE against representative one-stage frameworks (IPL [1], CPL [2]).
>
> The experimental results in Table 1 show that while these one-stage methods improve upon PT by bypassing explicit reward modeling, OPRIDE consistently and substantially outperforms them across nearly all tasks. This demonstrates the effectiveness of our iterative two-stage framework in identifying more informative queries for learning the reward function.
>
> We appreciate you highlighting these important baselines. We have expanded our discussion of one-stage frameworks in the Related Work section and included these new experimental results in Appendix E of the revised manuscript.
>
> | Tasks | PT | IPL | CPL | OPRIDE |
> |:-:|:-:|:-:|:-:|:-:|
> | lever-pull | 49.2$\pm$3.7 | 50.2$\pm$2.1 | 50.1$\pm$2.5 | 51.8$\pm$1.6|
> | peg-insert-side | 16.8$\pm$0.1 | 53.8$\pm$0.4 |  54.7$\pm$0.3 | 79.0$\pm$0.2 |
> |  plate-slide | 4.9$\pm$0.0 | 60.9$\pm$5.6 | 61.7$\pm$4.8 | 79.9$\pm$4.6 |
> | push | 16.7$\pm$5.0 | 38.5$\pm$4.2 | 39.2$\pm$3.9 | 59.1$\pm$5.4 |
> | push-back | 1.1$\pm$0.4 | 9.8$\pm$1.6 | 9.3$\pm$1.8 | 17.7$\pm$2.0 |
> | push-wall | 74.8$\pm$14.4 | 85.8$\pm$4.7 | 87.3$\pm$3.8 |102.2$\pm$1.2 |
> | reach | 82.0$\pm$0.8 | 84.2$\pm$0.2 | 84.7$\pm$0.6 |88.0$\pm$0.5 |
> | soccer | 51.3$\pm$4.1 | 52.3$\pm$4.7 | 53.6$\pm$3.9 | 45.4$\pm$3.9 |
> | sweep-into | 9.8$\pm$0.2 | 54.2$\pm$0.3 | 56.8$\pm$0.3 | 71.6$\pm$0.1 |
> | sweep | 8.0$\pm$0.4 | 69.3$\pm$1.1 | 68.2$\pm$1.3 | 78.5$\pm$1.0 |
>
> Table~1. Additional comparison with one-stage framework.
>
> **W2: Discussion CLARIFY [3] and LiRE [4] in the Related Works section.**
>
> **A for W2:**
> We thank you for this valuable suggestion. In the revised manuscript, we have incorporated a dedicated discussion of CLARIFY [3] and LiRE [4] into the Related Work section to better contextualize our contributions.
>
> **W3, W4, Q3: How would the proposed framework behave under noisy or inconsistent human feedback? Experiments on pixel-based domains.**
>
> **A for W3, W4, Q3:**
> We thank the reviewer for this excellent suggestion. To address these points, we conducted new experiments on pixel-based Atari environments, incorporating feedback from actual human evaluators to rigorously test OPRIDE's robustness. We recruited 15 experienced gamers who provided 50 pairwise comparisons each, using an interface populated by queries from OPRIDE.
>
> As shown in Table 2, OPRIDE achieves state-of-the-art performance across all five games. This strong result holds even with the dual challenges of high-dimensional visual inputs and the inherent noise of real human feedback, validating the robustness of our method. We attribute this success to two key factors: (1) our exploration strategy, which focuses on segment-level value differences, remains effective in the latent space of a CNN encoder; and (2) our variance-based discount scheduling (VDS) effectively mitigates over-optimization risks that are amplified by noisy human labels.
>
> These new results and a detailed description of the experimental setup have been added to Appendix E.
>
> |Task|OPRL|PT|PT+PDS|IDRL|OPRIDE|
> |:-:|:-:|:-:|:-:|:-:|:-:|
> |Pong|9.6$\pm$1.4|9.4$\pm$1.0|8.5$\pm$1.8|15.3$\pm$1.1|**17.8$\pm$1.3**|
> |Breakout|125.9$\pm$14.2|79.9$\pm$13.9|86.3$\pm$13.8|153.7$\pm$14.5|**256.7$\pm$14.3**|
> |Q*bert|7924.6$\pm$376.1|7482.9$\pm$353.4|6844.2$\pm$394.6|8294.1$\pm$359.7|**13535.2$\pm$327.2**|
> |Seaquest|2784.1$\pm$72.7|2538.3$\pm$78.9|2459.6$\pm$74.5|2941.2$\pm$69.2|**3478.4$\pm$71.3**|
> |Asterix|164.9$\pm$21.5|155.8$\pm$24.6|146.3$\pm$27.4|357.4$\pm$30.1|**426.9$\pm$28.7**|
>
> Table 2. Additional experiments on Atari tasks with human-in-the-loop.

---

> ### Author Response · Authors · 2025-11-21
> **Response to Reviewer 1iWt (II)**
>
> **Q1: How high variance in value estimation directly relates to overestimation?**
>
> **A for Q1:**
> To be precise, high variance in the value ensemble is an indicator of epistemic uncertainty, not a direct cause of overestimation. However, the two are strongly correlated in offline RL settings. Overly optimistic value estimates often arise from reward extrapolation in sparsely covered regions of the state-action space. Since each ensemble member may extrapolate differently based on spurious correlations in the data, these regions of overestimation manifest as high variance across the ensemble. Therefore, we use variance as a reliable and practical proxy to identify regions where overestimation is likely to occur.
>
> **Q2: How does the proposed VDS compare to these existing techniques?**
>
> **A for Q2:**
> Following your suggestion, we conducted an ablation study comparing our Variance-aware Discount Scheduling (VDS) to a common pessimism baseline, worst-case optimization (WCO), which uses the minimum prediction among ensemble reward models [5].
>
> The results in Table 5 show that VDS provides a substantial and consistent benefit over WCO. We hypothesize that simply taking the minimum over the reward ensemble is insufficient in our setting. Our dataset is dominated by random actions (95\%), with only a small fraction of perturbed trajectories. This high proportion of random data introduces a significant distribution shift relative to any competent policy, a condition known to exacerbate overestimation in offline RL. VDS more effectively counteracts this by directly regularizing the value function in uncertain regions, leading to much more stable and effective learning.
>
> |Tasks | OPRIDE (WCO) | OPRIDE (VDS)|
> |:-:|:-:|:-:|
> |bin-picking | 53.4$\pm$19.0 | 93.3$\pm$3.2|
> |button-press-wall | 59.4$\pm$0.9 | 77.7$\pm$0.1|
> |door-close | 62.4$\pm$8.7 | 94.8$\pm$1.1|
> |faucet-close | 46.2$\pm$0.2 |73.1$\pm$0.8|
> |peg-insert-side|12.4$\pm$1.4 |79.0$\pm$0.2|
> |reach | 84.3$\pm$0.9 | 88.0$\pm$0.5|
> |sweep | 8.0$\pm$0.1 |78.5$\pm$1.0
> Table 5. Ablation study about worst-case optimization and VDS.
>
> Thank you again for your thorough and insightful feedback, which has been instrumental in improving our work. We hope our responses and the additional results have fully addressed your concerns.
>
> **Reference**
>
> [1] Hejna, J., Sadigh, D. (2023). Inverse preference learning: Preference-based rl without a reward function. Advances in Neural Information Processing Systems, 36, 18806-18827.
>
> [2] Hejna, J., Rafailov, R., Sikchi, H., Finn, C., Niekum, S., Knox, W. B., Sadigh, D. Contrastive Preference Learning: Learning from Human Feedback without Reinforcement Learning. In The Twelfth International Conference on Learning Representations.
>
> [3] Mu, N., Hu, H., Hu, X., Yang, Y., XU, B., Jia, Q. S. CLARIFY: Contrastive Preference Reinforcement Learning for Untangling Ambiguous Queries. In Forty-second International Conference on Machine Learning.
>
> [4] Choi, H., Jung, S., Ahn, H., Moon, T. (2024, July). Listwise Reward Estimation for Offline Preference-based Reinforcement Learning. In International Conference on Machine Learning (pp. 8651-8671). PMLR.
>
> [5] Coste, T., Anwar, U., Kirk, R., Krueger, D. Reward Model Ensembles Help Mitigate Overoptimization. In The Twelfth International Conference on Learning Representations.

---

> > ### Comment · Reviewer_1iWt · 2025-11-25
> >
> > Thank you for the detailed responses. Most of my concerns have been adequately addressed. I would increase the score to 6 (from 4).

---

> > > ### Author Response · Authors · 2025-11-25
> > > **Thanks for raising the score!**
> > >
> > > We would like to thank the reviewer for raising the score! We also appreciate the valuable comments, which helped us significantly improve the paper's strengths. We will incorporate all feedback in the final version.

---

### Official Review · Reviewer_7G3h · 2025-10-29

**Soundness:** 3
**Presentation:** 3
**Contribution:** 3
**Rating:** 6
**Confidence:** 4

**Summary:**

This paper introduces OPRIDE, a novel algorithm designed to enhance query efficiency in offline PbRL. The authors demonstrate that OPRIDE achieves statistical efficiency by analyzing the suboptimality gap. Comprehensive experiments on Meta-World and D4RL benchmarks validate the superior performance of the proposed algorithm.

**Strengths:**

- The query efficiency is an important problem in offline PbRL.
- OPRIDE achieves SOTA in most tasks across Meta-World and Antmaze benchmarks.
- The proposed method is solid with theoretical guarantees.

**Weaknesses:**

- There are differences between the theoretical Algorithm 2 and the practical Algorithm 1; however, I fully understand that this simplification is necessary for theoretical analysis.

**Questions:**

- I'm intrigued by one observation: adding as little as 0.1% to 0.2% of additional data consistently improved performance without signs of leveling off. Could you clarify at what point this trend starts to plateau? For example, does the effect diminish around 1%, 3%, 5%, or even higher? At what point does adding more data stop producing meaningful gains? Additionally, could you provide any insight into why such a small fraction of data has such a significant impact?

Please include this experiment in the paper.

---

> ### Author Response · Authors · 2025-11-21
> **Response to Reviewer 7G3h (I)**
>
> Dear Reviewer,
>
> Thank you for your insightful questions. We appreciate the opportunity to clarify the connection between our theory and practice, and to provide a deeper analysis of our method's query efficiency. We believe the following responses and new experiments address your concerns and have strengthened the paper.
>
> **W1: The differences between the theoretical Algorithm 2 and the practical Algorithm 1.**
>
> **A for W1:**
> We deeply appreciate your understanding of such simplifications in our theoretical analysis.
>
> The core query selection mechanisms in Algorithm 1 (practical) and Algorithm 2 (theoretical) are identical; Equation (7) is a special case of Equation (12). The only difference lies in the discount factor adjustment mechanism during the policy extraction stage of Algorithm 1. This mechanism is introduced to mitigate the over-optimization problem that can arise from a poorly learned reward function.
>
> Consequently, the conclusions from our theoretical analysis apply directly to the query selection component of Algorithm 1. It is also worth noting that the adjustment of the discount factor is theoretically motivated. As demonstrated in prior work [1,2], a smaller discount factor yields a more pessimistic estimate of the value function, which is beneficial in offline settings.
>
> **Q1.1: What point the trend of improved performance starts to plateau?**
>
> **A for Q1.1:**
> As suggested, we conduct experiments with various query numbers.
> The experimental results in Table~1 show that performance gains accelerate most rapidly between 0.1\% and 0.2\% of queries. Beyond 0.3\%, improvements gradually taper off, ultimately stabilizing at around 1\%. At this point, adding further queries yields only marginal performance benefits.
> The new experimental results have been added to the Appendix E of the revised
> manuscript.
>
> |Query Number | 1 (BC) | 10 (0.1\%) | 13 | 20 (0.2\%) | 30 (0.3\%) | 50 (0.5\%) |100 (1.0\%) |
> |:-:|:-:|:-:|:-:|:-:|:-:|:-:|:-:|
> | Hammer-v2 | 13.2$\pm$3.3 | 50.3$\pm$3.2 | 55.4$\pm$2.6 | 75.9$\pm$2.2 | 79.1$\pm$2.6 | 79.5$\pm$2.7 | 79.9$\pm$2.3 |
> | Coffee-push-v2 | 2.5$\pm$1.1 | 56.7$\pm$2.8 | 57.6$\pm$2.5 | 55.3$\pm$3.1 | 58.9$\pm$3.7 | 59.4$\pm$3.5 | 59.5$\pm$3.4 |
> | Disassemble-v2 | 9.3$\pm$1.7 | 26.6$\pm$2.8 | 27.8 $\pm$ 3.1 | 30.6$\pm$5.2 | 35.7$\pm$4.8 | 37.2$\pm$4.6 | 38.3$\pm$5.1 |
> | Push-v2 | 1.2$\pm$0.6 | 50.9$\pm$1.8 | 63.7$\pm$2.1 | 76.8$\pm$2.4 | 82.7$\pm$2.6 | 83.6$\pm$2.3 | 84.0 $\pm$2.2 |
>
> Table 1. Performance of OPRIDE with various query numbers.

---

> ### Author Response · Authors · 2025-11-21
> **Response to Reviewer 7G3h (II)**
>
> **Q1.2: Why such a small fraction of data has such a significant impact?**
>
> **A for Q1.2:**
> We attribute this phenomenon to two factors: offline RL setting and efficient query mechanism.
>
> **Offline RL setting:**
> In the offline RL setting, the size and quality of unlabeled datasets remain unchanged since policy cannot explore new data.
> We train the reward function using a few queries, followed by dataset relabeling for training.
> Therefore, offline pbrl methods [3, 4] requires significantly fewer queries overall compared to the online pbrl [5, 6] counterpart.
>
> **Efficient query mechanism:**
> Our proposed query selection mechanism  (Equation 7) focuses only on queries that can improve the policy, disregarding those with minimal impact on policy improvement, significantly reducing our query requirements.
> We will conduct detailed explanation from three perspectives: intuition, theory, and experiments.
>
> - **Intuition:** Equation 7 aims to select queries that most effectively minimize the **diameter of the uncertainty set** for the value function. The diameter represents the maximum possible disagreement between any two candidate value functions on any two policies. Reducing this diameter is proportional to the information gain from a query. Therefore, minimizing this diameter is equivalent to maximizing the information gain for each query.
>
> - **Mathematics:** This objective is directly linked to the information ratio $\Gamma$ [7, 8]. The diameter of the uncertainty set is upper-bounded by:
> $$
> P\left(\text{diam}(\mathcal{R}) \leq \Gamma\_{\delta} \sqrt{I(\mathcal{R};\mathcal{D}\_{\text{query}})}\right) \geq 1-\delta
> $$
> where $\text{diam}(\mathcal{R}) = \max\_{R\_1,R\_2 \in \mathcal{R}} \max\_{\pi\_1,\pi\_2 \in \Pi} |(R\_1(\pi\_1)-R\_1(\pi\_2)) - (R\_2(\pi\_1)-R\_2(\pi\_2))|$, and $I(\mathcal{R};\mathcal{D}\_{\text{query}})$ is the mutual information between the reward function class $\mathcal{R}$ and the query dataset $\mathcal{D}\_{\text{query}}$. Maximizing the information gain $I$ directly corresponds to reducing the diameter.
>
> - **Comparison:** Maximizing information gain over the **optimal policy ($\pi^*$)** is more efficient than alternatives.
> Maximizing information gain over the **reward function ($r$)**, a common approach in PbRL, can be inefficient. It may lead the algorithm to focus on refining reward estimates in regions of the state space that are irrelevant to the optimal policy. Mathematically, the sample complexity of reward function estimation is proportional to the Eluder dimension of the reward function class, $d\_{\text{Elu}}(\mathcal{R})$, while our approach's complexity relates to the Eluder dimension of the optimal value function class. It is often the case that $d\_{\text{Elu}}(\mathcal{V}^*) \ll d\_{\text{Elu}}(\mathcal{R})$.
>
> - **Experiments:** We conducted an experiment comparing our query selection strategy with one based on reward function uncertainty. We replaced our IDE module with a disagreement-based module, termed OPRIDE (Disagreement). The results in Table 2 show that focusing on the optimal policy yields substantially better performance.
>
> | Tasks | OPRIDE (Disagreement) | OPRIDE (Original) |
> |:-:|:-:|:-:|
> | bin-picking | 78.5$\pm$17.8|**93.3$\pm$3.2** |
> | button-press-wall | 67.4$\pm$5.4 | **77.7$\pm$0.1** |
> | door-close | 88.3$\pm$0.7 | **94.8$\pm$1.1** |
> | faucet-close | 48.7$\pm$0.6 | **73.1$\pm$0.8** |
> | peg-insert-side | 9.7$\pm$8.5 | **79.0$\pm$0.2** |
> | reach | 86.6$\pm$0.1 | **88.0$\pm$0.5** |
> | sweep | 18.2$\pm$2.9 | **78.5$\pm$1.0** |
>
> Table~2. Ablation study for the query selection module on the Meta-World tasks.
>
> Thank you again for your valuable comments. We sincerely hope our additional experimental results and explanations have addressed your concerns.
>
>
> **Reference**
>
> [1] The dependence of effective planning horizon on model accuracy. Proceedings of the 2015 international conference on autonomous agents and multiagent systems. 2015.
>
> [2] On the role of discount factor in offline reinforcement learning. International conference on machine learning. PMLR, 2022.
>
> [3] Benchmarks and Algorithms for Offline Preference-Based Reward Learning. Transactions on Machine Learning Research.
>
> [4] Preference Transformer: Modeling Human Preferences using Transformers for RL. In The Eleventh International Conference on Learning Representations.
>
> [5] B-pref: Benchmarking preference-based reinforcement learning. arXiv preprint arXiv:2111.03026 (2021).
>
> [6] PEBBLE: Feedback-Efficient Interactive Reinforcement Learning via Relabeling Experience and Unsupervised Pre-training. In International Conference on Machine Learning (pp. 6152-6163). PMLR.
>
> [7] An information-theoretic analysis of thompson sampling. Journal of Machine Learning Research 17.68 (2016): 1-30.
>
> [8] Information-theoretic confidence bounds for reinforcement learning. Advances in neural information processing systems 32 (2019).

---

> > ### Comment · Reviewer_7G3h · 2025-11-26
> >
> > Thanks for your response. I have no other concerns and have decided to maintain my positive score.

---

> > > ### Author Response · Authors · 2025-11-26
> > > **Thanks for the positive score**
> > >
> > > We would like to thank the reviewer for standing by a positive score. We also appreciate the valuable comments, which helped us significantly improve the paper's strengths.

---

### Official Review · Reviewer_5bc5 · 2025-10-31

**Soundness:** 3
**Presentation:** 3
**Contribution:** 2
**Rating:** 4
**Confidence:** 4

**Summary:**

This paper proposes OPRIDE to improve query efficiency in offline PbRL through two main components: (1) in-dataset exploration that selects queries by maximizing value function disagreement between ensemble members, and (2) variance-based discount scheduling (VDS) that adjusts the discount factor based on ensemble prediction variance to mitigate reward overoptimization. The experimental evaluation demonstrates strong query efficiency with comprehensive ablation studies.

**Strengths:**

1. Proposed method demonstrates substantially higher query efficiency than prior work, achieving compelling results with only 10 queries across diverse domains.
2. The paper identifies and addresses reward overestimation in offline PBRL, an issue that has been relatively overlooked in previous studies.
3. The use of critic-based query sampling, instead of reward model-based sampling, is intuitive and directly targets policy-relevant queries.

**Weaknesses:**

1. Missing baselines: While the paper compares OPRIDE with methods such as OPRL and PT, it does not include more recent and orthogonal approaches that bypass reward modeling (IPL [1], DPPO [2], and CPL [3]) or enhance query sampling with sequential ranked lists (LiRE[4]). The lack of these comparisons weakens the experimental contribution.

2. Concerns about iterative learning: Unlike conventional two-stage offline PbRL methods that learn V and Q only once during policy extraction, OPRIDE iteratively trains V and Q during reward learning. This may introduce additional computational overhead compared to standard two-stage approaches. Furthermore, plasticity loss and capacity loss of neural networks may prevent both reward and corresponding value estimators from properly adapting to updated reward estimates as new preference data arrive [5, 6].

3. Limited experimental scope: All experiments rely on vector-based state inputs, which limits practical applicability in pixel-based environments. In addition, considering the focus of PbRL, the paper also should provide actual human-in-the-loop experiments.

4. Scalability concerns with mandatory ensembles: The method depends on ensemble reward models to compute value difference and measure variance. While ensemble can improve robustness, this requirement become impractical when reward models are very large, limiting scalability. The paper uses only M=2, but even this may be prohibitive in certain real-world settings.

[1] Hejna, J., & Sadigh, D. (2023). Inverse preference learning: Preference-based rl without a reward function. Advances in Neural Information Processing Systems, 36, 18806-18827.
[2] An, G., Lee, J., Zuo, X., Kosaka, N., Kim, K. M., & Song, H. O. (2023). Direct preference-based policy optimization without reward modeling. Advances in Neural Information Processing Systems, 36, 70247-70266.
[3] Hejna, J., Rafailov, R., Sikchi, H., Finn, C., Niekum, S., Knox, W. B., & Sadigh, D. Contrastive Preference Learning: Learning from Human Feedback without Reinforcement Learning. In The Twelfth International Conference on Learning Representations.
[4] Choi, H., Jung, S., Ahn, H., & Moon, T. (2024, July). Listwise Reward Estimation for Offline Preference-based Reinforcement Learning. In International Conference on Machine Learning (pp. 8651-8671). PMLR.
[5] Lyle, C., Rowland, M., & Dabney, W (2022). Understanding and Preventing Capacity Loss in Reinforcement Learning. In International Conference on Learning Representations.
[6] Lyle, C., Zheng, Z., Nikishin, E., Pires, B. A., Pascanu, R., & Dabney, W. (2023, July). Understanding plasticity in neural networks. In International Conference on Machine Learning (pp. 23190-23211). PMLR.

**Questions:**

1. Interpretation of survival instinct: The authors argue that using datasets with only ~5% perturbed trajectories makes their setup more challenging than [1]. However, this seems contradictory: survival instinct tends to strengthen when expert-like trajectories dominate, making learning easier under misspecified rewards. Thus, a low perturbation ratio likely makes the task easier, not harder. Clarifying this conceptual mismatch would improve the paper’s experimental claims.

2. Connection between Eluder dimension and proposed method: What is the specific connection between Eluder dimension in Theorem 4 and the practical algorithm (Algorithm 1)?

3. Variance and overestimation relationship: How does high variance in value estimation directly lead to overestimation? Overestimation is typically measured as the gap between predicted Q-values and actual discounted sum of rewards under the same reward function, or judged via expected Q-values. High variance reflects uncertainty but does not necessarily imply overestimation.

4. Discount factor adjustment and overestimation: How does adjusting the discount factor in VDS specifically mitigate overestimation? Section 3.2 refers to pessimism, but the mechanism is unclear. Reducing the discount factor may induce more myopic value estimates rather than truly mitigating overestimation. Does this adjustment simply reduce horizon length, or is there a more fundamental connection?

5. Heuristic nature of m% threshold: Using a static threshold (Top m%) to identify high-variance samples for discount adjustment is quite heuristic. What principled improvements could be made?

6. Scalability concern (related to Weakness 4) : What solutions exist when for cases where only a single reward model can be used due to computational constraints?

[1] Li, A., Misra, D., Kolobov, A., & Cheng, C. A. (2023). Survival instinct in offline reinforcement learning. Advances in neural information processing systems, 36, 62062-62120.

---

> ### Author Response · Authors · 2025-11-21
> **Response to Reviewer 5bc5 (I)**
>
> Dear Reviewer,
>
> Thank you for your valuable and constructive comments. We have performed additional experiments and analysis to address your concerns, and we believe the following responses have strengthened our manuscript.
>
> **W1: Comparison with one-stage framework and sequential ranked list methods.**
>
> **A for W1:**
> As suggested, we compare OPRIDE with one-stage framework (IPL, CPL) [1,2] and sequential ranked list method (LiRE) [3].
> The experimental results in Table 1 show that while these baselines improve upon PT by either bypassing explicit reward modeling or using enhanced query sampling, OPRIDE consistently outperforms them across nearly all tasks. This demonstrates the effectiveness of our iterative two-stage framework in identifying more informative queries for learning the reward function.
>
> We appreciate you highlighting these important baselines. We have expanded our discussion of them in the Related Work section and included these new experimental results in Appendix E of the revised
>
> | Tasks | PT | IPL | CPL | LiRE | OPRIDE |
> |:-:|:-:|:-:|:-:|:-:|:-:|
> | lever-pull | 49.2$\pm$3.7 | 50.2$\pm$2.1 | 50.1$\pm$2.5 | 51.2$\pm$1.8 | 51.8$\pm$1.6|
> | peg-insert-side | 16.8$\pm$0.1 | 53.8$\pm$0.4 |  54.7$\pm$0.3 | 63.1$\pm$0.2 | 79.0$\pm$0.2 |
> |  plate-slide | 4.9$\pm$0.0 | 60.9$\pm$5.6 | 61.7$\pm$4.8 | 68.3$\pm$4.3 | 79.9$\pm$4.6 |
> | push | 16.7$\pm$5.0 | 38.5$\pm$4.2 | 39.2$\pm$3.9 | 45.5$\pm$3.7 | 59.1$\pm$5.4 |
> | push-back | 1.1$\pm$0.4 | 9.8$\pm$1.6 | 9.3$\pm$1.8 | 13.2$\pm$1.5 | 17.7$\pm$2.0 |
> | push-wall | 74.8$\pm$14.4 | 85.8$\pm$4.7 | 87.3$\pm$3.8 | 90.2$\pm$3.6 |102.2$\pm$1.2 |
> | reach | 82.0$\pm$0.8 | 84.2$\pm$0.2 | 84.7$\pm$0.6 | 85.4$\pm$0.3 |88.0$\pm$0.5 |
> | soccer | 51.3$\pm$4.1 | 52.3$\pm$4.7 | 53.6$\pm$3.9 | 54.2$\pm$3.6 | 45.4$\pm$3.9 |
> | sweep-into | 9.8$\pm$0.2 | 54.2$\pm$0.3 | 56.8$\pm$0.3 | 62.9$\pm$0.2 | 71.6$\pm$0.1 |
> | sweep | 8.0$\pm$0.4 | 69.3$\pm$1.1 | 68.2$\pm$1.3 | 71.3$\pm$1.6 | 78.5$\pm$1.0 |
>
> Table~1. Additional comparison with one-stage framework and sequential ranked list methods.
>
> **W2: Iterative learning introduces additional computational overhead.
> Plasticity loss and capacity loss of neural networks may prevent properly adapting to updated reward estimates.**
>
> **A for W2:**
> We appreciate these thoughtful points. We address these two distinct concerns below:
>
> - **Computational overhead:** While our iterative approach differs from conventional two-stage methods, we posit that this design is crucial for maximizing information gain from a limited query budget. Regarding the computational cost, our use of multi-head ensembles with shared backbones makes the process highly efficient. As shown in Table 12 (Appendix E), the training time scales sub-linearly with the ensemble size (e.g., a marginal increase of only 0.1 to 0.3 hours when scaling M from 2 to 10).
>
> - **Plasticity and capacity loss:** We acknowledge that iterative updates can, in theory, lead to catastrophic forgetting or capacity saturation. To mitigate this, our ensemble-based reward model maintains diversity through independently initialized heads, which helps preserve plasticity. In the revised manuscript, we have incorporated this discussion into a new Limitations section and identified the explicit management of plasticity as a promising direction for future work.
>
>
> **W3: Pixel-based and actual human-in-the-loop environments.**
>
> **A for W3:**
> We thank the reviewer for this excellent suggestion. We have conducted new experiments on five pixel-based Atari environments, incorporating feedback from actual human evaluators to rigorously test OPRIDE's generalization. We recruited 15 experienced gamers who provided 50 pairwise comparisons each, using an interface populated by queries from OPRIDE.
>
> As shown in Table 2, OPRIDE achieves state-of-the-art performance across all five games. This strong result holds even with the dual challenges of high-dimensional visual inputs and the inherent noise of human feedback, validating the robustness of our method. We attribute this success to two factors: (1) our exploration strategy, which focuses on segment-level value differences, remains effective in the latent space of a CNN encoder; and (2) our variance-based discount scheduling effectively mitigates over-optimization on noisy human labels.
>
> These new results and a detailed description of the experimental setup have been added to Appendix E.
>
> |Task|OPRL|PT|PT+PDS|IDRL|OPRIDE|
> |:-:|:-:|:-:|:-:|:-:|:-:|
> |Pong|9.6$\pm$1.4|9.4$\pm$1.0|8.5$\pm$1.8|15.3$\pm$1.1|**17.8$\pm$1.3**|
> |Breakout|125.9$\pm$14.2|79.9$\pm$13.9|86.3$\pm$13.8|153.7$\pm$14.5|**256.7$\pm$14.3**|
> |Q*bert|7924.6$\pm$376.1|7482.9$\pm$353.4|6844.2$\pm$394.6|8294.1$\pm$359.7|**13535.2$\pm$327.2**|
> |Seaquest|2784.1$\pm$72.7|2538.3$\pm$78.9|2459.6$\pm$74.5|2941.2$\pm$69.2|**3478.4$\pm$71.3**|
> |Asterix|164.9$\pm$21.5|155.8$\pm$24.6|146.3$\pm$27.4|357.4$\pm$30.1|**426.9$\pm$28.7**|
>
> Table 2. Additional experiments on Atari tasks with human-in-the-loop.

---

> ### Author Response · Authors · 2025-11-21
> **Response to Reviewer 5bc5 (II)**
>
> **W4 and Q6: Ensemble reward models.**
>
> **A for W4 and Q6:**
> We agree that large ensembles can be resource-intensive. We wish to clarify that while ensembles are an effective method for uncertainty quantification, they are not a mandatory component of our framework. The core of OPRIDE's exploration strategy—maximizing segment-value differences—is compatible with any valid uncertainty measure. To address computational constraints, OPRIDE can readily incorporate more lightweight alternatives, such as Monte Carlo dropout or last-layer ensembles, which provide effective variance estimation with a single base model.
>
> **Q1: Interpretation of survival instinct.**
>
> **A for Q1:**
> Thank you for the opportunity to clarify this point.  Our dataset is, in fact, more challenging than that of Survival Instinct.  As detailed in Appendix F, our dataset is predominantly composed of random trajectories (95\%), with only a small fraction (5\%) of perturbed trajectories.  This contrasts with the Survival Instinct dataset, which consists entirely of perturbed expert trajectories.
>
> **Q2: What is the specific connection between Eluder dimension in Theorem 4 and the practical algorithm (Algorithm 1)?**
>
> **A for Q2:**
> The Eluder dimension is a theoretical complexity measure that is instrumental in our analysis. Specifically, as shown in Theorem 4, the suboptimality bound of OPRIDE (Equation 13) directly depends on the Eluder dimension of the value function class via the preference error term. A smaller Eluder dimension implies that the function class is "easier" to learn from limited data, leading to a tighter regret bound for our algorithm.
>
> **Q3: How does high variance in value estimation directly lead to overestimation?**
>
> **A for Q3:**
> To be precise, high variance in the value ensemble is an indicator of epistemic uncertainty, not a direct cause of overestimation.  However, the two are strongly correlated in offline RL settings.  Overly optimistic value estimates often arise from reward extrapolation in sparsely covered regions of the state-action space.  Since each ensemble member may extrapolate differently based on spurious correlations in the data, these regions of overestimation manifest as high variance across the ensemble.  Therefore, we use variance as a reliable and practical proxy to identify regions where overestimation is likely to occur.
>
> **Q4: How does adjusting the discount factor in VDS specifically mitigate overestimation?**
>
> **A for Q4:**
> Our approach is motivated by both theory and empirical results.
>
> - Theoretically, using a pessimistic or smaller discount factor is a well-established principle in offline RL for ensuring robust policy learning under uncertainty [4, 5]. It effectively regularizes the policy by shrinking the effective planning horizon in uncertain regions, thus preventing the exploitation of erroneously high long-term values.
> - Empirically, our ablation study (Table 3) confirms the necessity of VDS. Removing it leads to a significant performance degradation across all tasks, highlighting its critical role in stabilizing the learning process.
>
> |Task|OPRIDE (VDS)|OPRIDE w/o VDS|
> |:-:|:-:|:-:|
> |bin-picking|93.3$\pm$3.2| 51.2$\pm$12.0|
> |button-press-wall|77.7$\pm$0.1|56.4$\pm$1.3|
> |door-close|94.8$\pm$1.1|60.7$\pm$5.8|
> |faucet-close|73.1$\pm$0.8| 43.9$\pm$2.1|
> |peg-insert-side|79.0$\pm$0.2| 10.6$\pm$2.1|
> |reach|88.0$\pm$0.5|82.7$\pm$2.5|
> |sweep|78.5$\pm$1.0| 19.2$\pm$1.1|
>
> Table 3. Ablation study for the discount scheduling (VDS).

---

> ### Author Response · Authors · 2025-11-21
> **Response to Reviewer 5bc5 (III)**
>
> **Q5: What principled improvements could be made for discount adjustment?**
>
> **A for Q5:**
> This is an excellent question. Following your suggestion, we investigated a more adaptive, continuous discount mechanism as an alternative to our threshold-based approach.
> Specifically, we implemented a continuous annealing strategy where the smaller discount factor, $\gamma\_{\text{small}}$, is adjusted based on the variance of the ensemble's value function estimates, $\text{Var}[Q\_{\phi_i}(s,a)]\_{i=1}^M$. In this setup, $\gamma\_{\text{small}}$ decreases as the variance increases, governed by the formula:
> $$
> \gamma\_{\text{small}} = \frac{\gamma}{\max(1, \alpha \cdot \text{Var}[Q\_{\phi\_i}(s,a)]\_{i=1}^M)}.
> $$
> The experimental results, presented in Table~4, show that the performance of this continuous annealing approach is comparable to our hard-penalty method.
>
> We can employ such softer confidence discount mechanism to mitigate the effects of nonstationarity in learning dynamics when variance estimates fluctuate across batches.
>
> | Tasks | Continuous annealing | Threshold-based |
> |:-:|:-:|:-:|
> | bin-picking | 94.1$\pm$3.7|93.3$\pm$3.2 |
> | button-press-wall | 77.4$\pm$0.3 | 77.7$\pm$0.1 |
> | door-close | 94.3$\pm$1.7 | 94.8$\pm$1.1 |
> | faucet-close | 71.6$\pm$0.9 | 73.1$\pm$0.8 |
> | peg-insert-side | 80.5$\pm$0.3 | 79.0$\pm$0.2 |
> | reach | 87.5$\pm$0.6 | 88.0$\pm$0.5 |
> | sweep | 79.8$\pm$1.2 | 78.5$\pm$1.0 |
>
> Table 4. Experiments for the adaptive, continuous confidence discount mechanism.
>
> Thank you again for your thorough and insightful feedback, which has been instrumental in improving our work. We hope our responses and the additional results have fully addressed your concerns.
>
> **Reference**
>
> [1] Hejna, J., Sadigh, D. (2023). Inverse preference learning: Preference-based rl without a reward function. Advances in Neural Information Processing Systems, 36, 18806-18827.
>
> [2] Hejna, J., Rafailov, R., Sikchi, H., Finn, C., Niekum, S., Knox, W. B., Sadigh, D. Contrastive Preference Learning: Learning from Human Feedback without Reinforcement Learning. In The Twelfth International Conference on Learning Representations.
>
> [3] Choi, H., Jung, S., Ahn, H., Moon, T. (2024, July). Listwise Reward Estimation for Offline Preference-based Reinforcement Learning. In International Conference on Machine Learning (pp. 8651-8671). PMLR.
>
> [4] Jiang, Nan, et al. "The dependence of effective planning horizon on model accuracy." Proceedings of the 2015 international conference on autonomous agents and multiagent systems. 2015.
>
> [5] Hu, Hao, et al. "On the role of discount factor in offline reinforcement learning." International conference on machine learning. PMLR, 2022.

---

> ### Author Response · Authors · 2025-11-27
> **Looking forward to further discussions!**
>
> Dear reviewer,
>
> We were wondering if our response and revision have cleared all your concerns. In the previous response, we have tried to address all the points you have raised. We would appreciate it if you could kindly let us know whether you have any other questions, so that we can still have time to respond and address. We are looking forward to discussions that can further improve our current manuscript. Thanks!
>
> Best regards,
>
> The Authors

---

> > ### Comment · Reviewer_5bc5 · 2025-11-28
> >
> > The clarification provided in the responses to W4 and Q6 indicates that large ensembles are not strictly required and that lighter-weight uncertainty estimation methods (e.g., Monte Carlo dropout, last-layer ensembles) can be adopted to reduce computational overhead. However, this statement appears to be in tension with the explanation given in the response to Q3, where reliable variance estimation is described as central to identifying regions of overestimation and mitigating epistemic uncertainty in offline RL.
> >
> > Given that OPRIDE’s exploration strategy relies on detecting segments with high epistemic uncertainty—operationalized via ensemble variance—it is not immediately clear whether the proposed lightweight alternatives can produce uncertainty estimates of comparable quality. Intuitively, the reduced model diversity of Monte Carlo dropout or last-layer ensembles might lead to under-dispersed uncertainty estimates, potentially making it harder to detect overestimated regions.
> >
> > To strengthen the authors’ claim about the generality and flexibility of their framework, it would be helpful to:
> > 	1.	Provide empirical results demonstrating that OPRIDE’s performance remains stable when using lighter uncertainty estimators, or
> > 	2.	Clarify theoretically or conceptually how sensitive OPRIDE is to the fidelity of the uncertainty measure, particularly in the context of value overestimation.
> >
> > Such clarification would improve the overall coherence of the paper and help assess whether OPRIDE can truly retain its performance benefits without full ensemble-based uncertainty estimation.

---

> ### Author Response · Authors · 2025-11-30
> **Further Response to Reviewer 5bc5**
>
> Dear Reviewer,
>
> Thank you for this insightful question. We have conducted additional experiments to address this point directly.
>
> - **Experiments for using lighter uncertainty estimators:** The core requirement for our exploration strategy is a mechanism to identify trajectories where the current policy is most uncertain. While a full multi-model ensemble is a robust way to estimate this uncertainty, it is not the only effective approach. Our new empirical results strongly support this claim. As shown in Table 1, when we replace the full ensembles with a more computationally efficient last-layer ensemble (which uses a single network backbone with multiple output heads), the performance remains nearly identical to the original OPRIDE. This minimal performance difference demonstrates that the quality of the uncertainty estimate is well-preserved, and the reduced architectural diversity of a last-layer ensemble does not significantly impair its ability to identify regions of high uncertainty for querying.
>
> - **Theoretical Analysis:** Theoretically, OPRIDE is not critically sensitive to the exact dispersion of the uncertainty estimate, provided that it can reliably distinguish between high and low-uncertainty states. The primary function of the policy variance is to serve as a selection criterion for informative queries, not as a perfectly calibrated absolute quantity. Therefore, even if more efficient estimators like last-layer ensembles or Monte Carlo dropout are slightly under-dispersed compared to full ensembles, they still provide a valid ranking of trajectories. This is because our theoretical analysis in Section 4 shows that the exploration strategy's efficiency depends on reducing the diameter of the policy uncertainty set, a goal that can be achieved as long as the uncertainty estimates are consistent and correlate well with the true policy error.
>
> Thanks again for your reply and we sincerely hope our further response has solved your remaining concerns.
>
> | Tasks | OPRIDE (Last-layer Ensembles) | OPRIDE |
> |:-:|:-:|:-:|
> | bin-picking | 93.0$\pm$2.7|93.3$\pm$3.2 |
> | button-press-wall | 76.8$\pm$0.5 | 77.7$\pm$0.1 |
> | door-close | 95.1$\pm$1.6 | 94.8$\pm$1.1 |
> | faucet-close | 72.8$\pm$0.7 | 73.1$\pm$0.8 |
> | peg-insert-side | 79.5$\pm$0.4 | 79.0$\pm$0.2 |
> | reach | 88.1$\pm$0.6 | 88.0$\pm$0.5 |
> | sweep | 77.9$\pm$1.1 | 78.5$\pm$1.0 |
>
> Table 1. Experiments for using lighter uncertainty estimators.

---

### Official Review · Reviewer_qmux · 2025-11-03

**Soundness:** 3
**Presentation:** 3
**Contribution:** 2
**Rating:** 6
**Confidence:** 2

**Summary:**

This paper proposes Offline PbRL via In-Dataset Exploration (OPRIDE), a novel algorithm designed to systematically enhance the query efficiency of offline PbRL. OPRIDE introduces a principled exploration strategy that identifies the most informative queries by analyzing value differences between trajectories, ensuring that each query maximally contributes to learning the optimal policy. Additionally, to prevent overoptimization of the learned reward function, particularly in regions with high uncertainty, OPRIDE incorporate a discount factor scheduling mechanism that dynamically adjusts the discount based on the variance in the reward estimation. Experiments are performed on diverse locomotion and manipulation tasks, including AntMaze and Meta-World. Results compared to existing baselines demonstrates improvements in query efficiency.

**Strengths:**

1. The method shifts query generation from purely reward-based uncertainty to policy-level uncertainty, focusing on the difference of value differences. This provides a more direct way to improve policy-relevant information efficiency compared to standard reward uncertainty sampling.
2. Using multiple reward and value function estimates improves robustness and helps quantify epistemic uncertainty in offline settings where exploration is limited.
3. Dynamically adjusting the discount factor based on value variance provides a practical, adaptive mechanism to mitigate reward overestimation and stabilize learning in noisy preference data.
4. Authors conducted comprehensive experimental evaluations, which demonstrate overall significant improvements compared to baselines.

**Weaknesses:**

1. Potential instability: Adaptive discount factor modulation could introduce nonstationarity in learning dynamics, especially when variance estimates fluctuate across batches.

**Questions:**

None

---

> ### Author Response · Authors · 2025-11-21
> **Response to Reviewer qmux**
>
> Dear Reviewer,
>
> Thank you for your thorough review and insightful comments. We have conducted additional experiments to address your concerns and believe the following clarifications have strengthened the paper.
>
> **W1: Adaptive discount factor modulation.**
>
> **A for W1:**
> We appreciate this constructive suggestion. To address your concern regarding nonstationarity in the learning dynamics, we investigated a more adaptive, continuous discount mechanism as an alternative to our proposed threshold-based approach. Specifically, we implemented a continuous annealing strategy where the smaller discount factor, $\gamma\_{\text{small}}$, is adjusted based on the variance of the ensemble's value function estimates, $\text{Var}[Q\_{\phi_i}(s,a)]\_{i=1}^M$. In this setup, $\gamma\_{\text{small}}$ decreases as the variance increases, governed by the formula:
> $$
> \gamma\_{\text{small}} = \frac{\gamma}{\max(1, \alpha \cdot \text{Var}[Q\_{\phi\_i}(s,a)]\_{i=1}^M)}.
> $$
> The experimental results, presented in Table 1, show that the performance of this continuous annealing approach is comparable to our hard-penalty method.
>
> We can employ such softer confidence discount mechanism to mitigate the effects of nonstationarity in learning dynamics when variance estimates fluctuate across batches.
>
> | Tasks | Continuous annealing | Threshold-based |
> |:-:|:-:|:-:|
> | bin-picking | 94.1$\pm$3.7|93.3$\pm$3.2 |
> | button-press-wall | 77.4$\pm$0.3 | 77.7$\pm$0.1 |
> | door-close | 94.3$\pm$1.7 | 94.8$\pm$1.1 |
> | faucet-close | 71.6$\pm$0.9 | 73.1$\pm$0.8 |
> | peg-insert-side | 80.5$\pm$0.3 | 79.0$\pm$0.2 |
> | reach | 87.5$\pm$0.6 | 88.0$\pm$0.5 |
> | sweep | 79.8$\pm$1.2 | 78.5$\pm$1.0 |
>
> Table 1. Ablation study for the discount factor mechanism on the Meta-World tasks.
>
> We sincerely thank you again for your timely and valuable feedback. Your constructive suggestions have been instrumental in improving the quality of this paper. We hope our responses have adequately addressed your concerns.

---

> ### Author Response · Authors · 2025-11-27
> **Looking forward to further discussions!**
>
> Dear reviewer,
>
> We were wondering if our response and revision have cleared all your concerns. In the previous response, we have tried to address all the points you have raised. We would appreciate it if you could kindly let us know whether you have any other questions, so that we can still have time to respond and address. We are looking forward to discussions that can further improve our current manuscript. Thanks!
>
> Best regards,
>
> The Authors

---

### Meta-Review · Area_Chair_fxTr · 2026-01-06

**Summary:**

This work proposes an exploration strategy to improve the efficiency of offline preference-based reinforcement learning. The main idea is to select informative queries based on value differences. The authors demonstrate the effectiveness of the proposed method in simulated environments. Reviewers initially raised concerns regarding baseline comparisons and evaluation across diverse environments. The authors have addressed several of these points, strengthening the empirical section and clarifying the methodology. While some issues remain open (see `Reviewer Concerns`), they do not detract from the core contribution. I encourage the authors to carefully reflect on the remaining reviewer comments and, where possible, further resolve them in the camera-ready version.

**Reviewer Concerns:**

### Reviewer qmux
* [resolved] Potential instability

### Reviewer 1iWt
* [partially resolved] Comparison with recent work (e.g., IPL and CPL): new experiments added. For some tasks, these baselines outperform the proposed method.
* [resolved] Discussion on two-stage offline PbRL
* [remaining] Experiments with non-ideal and noisy annotators (real humans): not addressed
* [resolved] Experiments with pixel-based environments: new Atari experiments added

### Reviewer 5bc5
* [partially resolved] Missing baselines
* [resolved] Concerns about iterative learning
* [resolved] Limited experimental scope
* [partially resolved] Scalability concerns

### Reviewer 7G3h
* [resolved] Differences between the theoretical Algorithm 2 and the practical Algorithm 1

**Reviewer Scores:**

* Reviewer qmux: 6 $\rightarrow$ 6

* Reviewer 5bc5: 4 $\rightarrow$ 4

* Reviewer 7G3h: 6 $\rightarrow$ 6

* Reviewer 1iWt: 4 $\rightarrow$ 6

---

### Decision · Program_Chairs · 2026-01-26

Accept (Poster)